# Novel long-range inhibitory nNOS-expressing hippocampal cells

**Zoé Christenson Wick[1]\*, Madison R Tetzlaff[2], Esther Krook-Magnuson[2]**

[1]Graduate Program in Neuroscience, University of Minnesota, Minneapolis, United States; [2]Neuroscience Department, University of Minnesota, Minneapolis, United States

**Abstract** The hippocampus, a brain region that is important for spatial navigation and episodic memory, benefits from a rich diversity of neuronal cell-types. Through the use of an intersectional genetic viral vector approach in mice, we report novel hippocampal neurons which we refer to as LINCs, as they are long-range inhibitory neuronal nitric oxide synthase (nNOS)-expressing cells. LINCs project to several extrahippocampal regions including the tenia tecta, diagonal band, and retromammillary nucleus, but also broadly target local CA1 cells. LINCs are thus both interneurons and projection neurons. LINCs display regular spiking non-pyramidal firing patterns, are primarily located in the stratum oriens or pyramidale, have sparsely spiny dendrites, and do not typically express somatostatin, VIP, or the muscarinic acetylcholine receptor M2. We further demonstrate that LINCs can strongly influence hippocampal function and oscillations, including interregional coherence. The identification and characterization of these novel cells advances our basic understanding of both hippocampal circuitry and neuronal diversity.

DOI: https://doi.org/10.7554/eLife.46816.001

## Introduction

The hippocampus is one of the most extensively studied brain regions (*Andersen et al., 2007*), and in CA1 alone, more than 20 types of inhibitory neurons have been previously described (*Freund and Buzsáki, 1996*; *Klausberger and Somogyi, 2008*). Each population of neurons plays a unique role in the circuitry (*Bezaire et al., 2016*; *Pelkey et al., 2017*; *Roux and Buzsáki, 2015*; *Soltesz, 2006*), and together, they allow for the emergent functionality of the hippocampus, including effective navigation through time and space (*Eichenbaum, 2014*) and the formation of episodic memories (*Lisman et al., 2017*). The hippocampus does not work in isolation and has extensive connections with other brain regions. Oscillations, and their synchrony or coherence, are believed to play an important role in coordinating the activity between the hippocampus and downstream regions (*Buzsáki et al., 2013*; *Colgin, 2011*; *Sirota et al., 2008*).

Despite extensive prior investigation of the neuronal populations in CA1 (reviewed by *Freund and Buzsáki, 1996*, *Klausberger and Somogyi, 2008* and *Pelkey et al., 2017*), recent work highlights that some cell types still lack proper characterization (*Harris et al., 2018*). Here, we report a novel population of cells which 1) are GABAergic, 2) express neuronal nitric oxide synthase (nNOS), and 3) have both local and long-range axonal projections. Therefore, on the basis of these unifying features, we refer to these cells as LINCs: long-range inhibitory nNOS-expressing cells. Although LINCs express nNOS and possess long-range axons, they do not appear to be simply hippocampal versions of cortical NOS-type I cells, nor do they closely match any other previously characterized hippocampal cell population, as detailed in depth below. In all, despite being relatively few in number, the properties of LINCs suggest that they can have a surprisingly robust impact on hippocampal network function, oscillatory dynamics, and inter-regional coherence.

**\*For correspondence:**
chri3433@umn.edu

**Competing interests:** The authors declare that no competing interests exist.

## Results

### Intersectional vector approach labels LINCs

The ability to identify, characterize, and manipulate LINCs rests on the use of a recently developed Cre- and Flp-dependent virus for the expression of eYFP-tagged ChR2 (AAV-DJ-hSyn-Con/Fon-hChR2(H134R)-eYFP-WPRE; *Fenno et al., 2014*) and on mice that express Cre in their nNOS+ neurons and Flpe in their Dlx5/6+ GABAergic cells (*Figure 1a,b*). This approach limits the expression of eYFP-tagged ChR2 to nNOS-expressing interneurons (*Figure 1c*), although nNOS-expression was difficult to detect in some labeled neurons and not all nNOS-immunopositive cells expressed eYFP (*Figure 1—figure supplement 1*). Although the eYFP-expressing cells were nNOS-immunopositive (as expected), they had unexpected morphologies, with broad, sparsely spiny dendrites (*Figure 1d*); we refer to these cells as LINCs. Viral injection into animals that were negative for Cre and/or Flpe did not result in opsin expression (*Figure 1e*), further confirming the specificity of the approach.

LINCs could be labeled with a viral injection targeting either the dorsal or ventral CA1 stratum oriens, with virally labeled cells found along the anterior-posterior extent of the hippocampus at a considerable distance from the site of injection (*Figure 2a*; *Figure 2—figure supplement 1*), suggestive of widespread processes. The somata of labeled LINCs were located primarily in the stratum

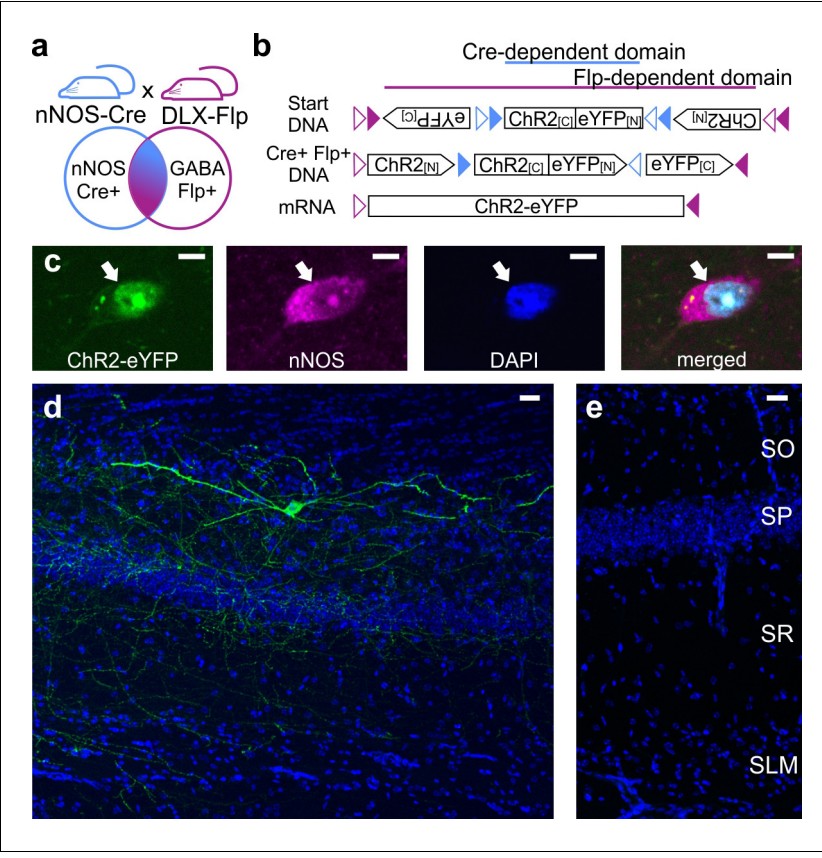

**Figure 1.** Intersectional genetic viral vector approach to target and characterize LINCs in the hippocampus. nNOS-Cre x Dlx5/6-Flpe mice (**a**) that are injected with an intersectional AAV-DJ-ChR2-eYFP vector (**b**) show selective expression in nNOS-positive (**c**) inhibitory neurons (**d**) (LINCs). No expression is seen in injected negative control animals (**e**). Panel (**b**) is based on *Fenno et al. (2014)*. Stratum oriens (SO), stratum pyramidale (SP), stratum radiatum (SR), stratum lacunosum moleculare (SLM). Scale bars: 5 μm (**c**); 25 μm (**d, e**).

DOI: https://doi.org/10.7554/eLife.46816.002

The following figure supplement is available for figure 1:

**Figure supplement 1.** LINCs express nNOS.

DOI: https://doi.org/10.7554/eLife.46816.003

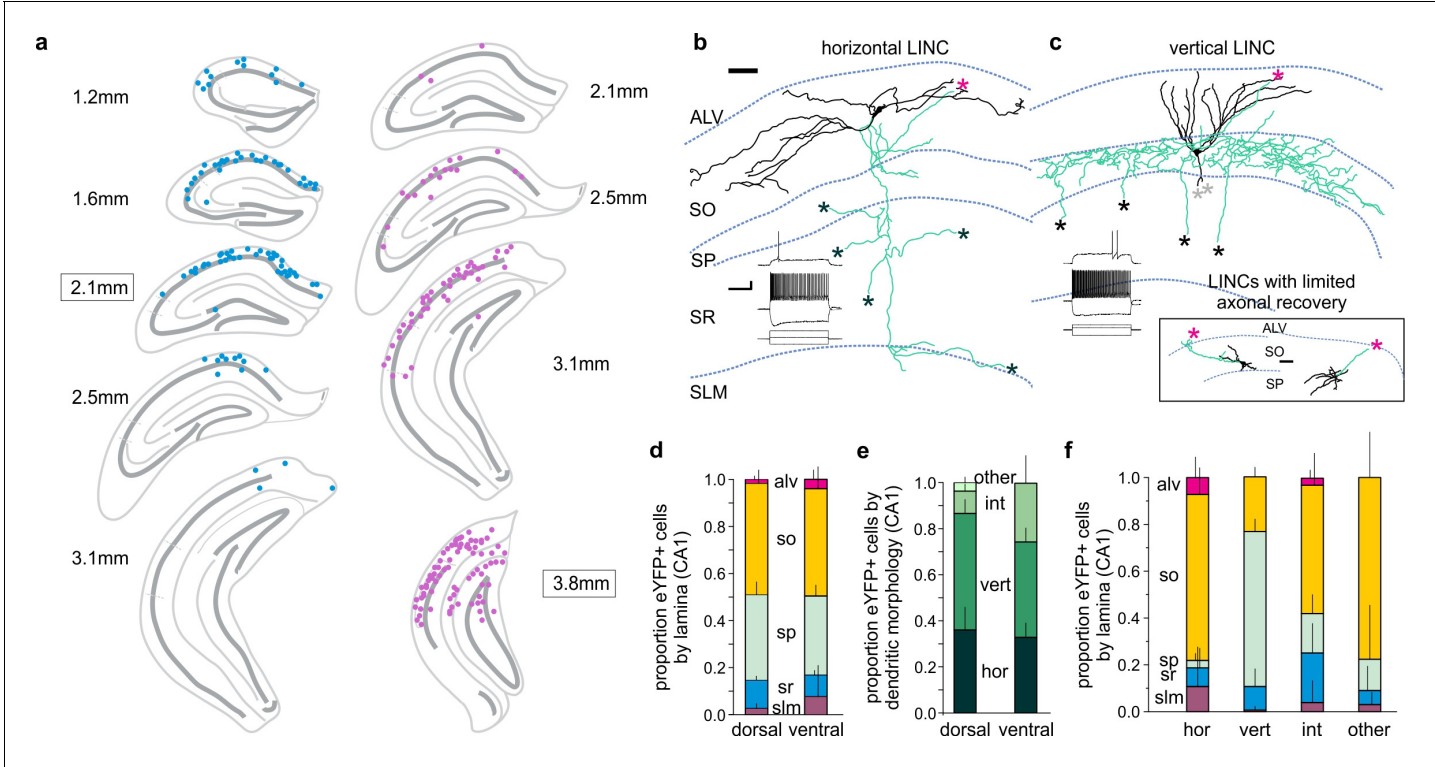

**Figure 2.** CA1 LINCs are typically located in SO or SP, largely display vertical or horizontal dendritic morphologies, and have axonal arbors consistent with long-range projections. (a) eYFP+ somata location from a 1-in-4 coronal series after dorsal (blue) or more ventral (violet) virus injection (approximate injection position outlined). (b–c) LINCs have largely horizontal (b) or vertical (c) dendrites (black). In addition to local axons (green), all filled LINCs showed a severed axon (asterisk), often en route to the alveus (magenta asterisks). These axons were present even with limited axonal recovery (inset box), and are suggestive of long-range projections. Note that the example LINC in (c) was found to be PV-immunonegative. Insets in panels (b, c): firing properties of the cells. (d–f) CA1 LINCs by lamina (d), dendritic morphology (e), and location of cell body (f). Horizontal (hor), vertical (vert), intermediate (int), or other dendritic morphologies were counted. Cells without discernable dendrites were included in 'other.' Dorsal and ventral refer to the location of the viral injection. Alveus (ALV), stratum oriens (SO), stratum pyramidale (SP), stratum radiatum (SR), stratum lacunosum moleculare (SLM). Data are displayed as mean + SD in (d–f). Scale bars (b, c): 50 µm; 20 mV, 500 pA and 200 ms.

DOI: https://doi.org/10.7554/eLife.46816.004

The following figure supplements are available for figure 2:

**Figure supplement 1.** LINCs are found in anterior, intermediate, and posterior regions of the hippocampus.

DOI: https://doi.org/10.7554/eLife.46816.005

**Figure supplement 2.** Labeling of putative neurogliaform cells depends, at least in part, on the location of virus injection.

DOI: https://doi.org/10.7554/eLife.46816.006

**Figure supplement 3.** Additional examples of LINC morphology.

DOI: https://doi.org/10.7554/eLife.46816.007

oriens or stratum pyramidale of CA1 (*Figure 2a*; *Figure 2—figure supplement 1*), and typically had either horizontally or vertically oriented dendrites (hLINC and vLINC, respectively) (*Figure 2b–f*). The dendritic morphology roughly corresponded to the somatic location: hLINCs were found primarily in the stratum oriens (SO), whereas vLINCs were found primarily in the stratum pyramidale (SP) (*Figure 2f*).

Previous literature in wild-type animals notes the existence of nNOS-immunopositive cells in stratum oriens that have 'largely horizontal dendrites', as well as occasional 'bitufted' cells in stratum pyramidale, but any further characterization of these cells was lacking (*Freund and Buzsáki, 1996*). However, we were still surprised that our intersectional approach labeled LINCs, and not other, better characterized, populations of nNOS-expressing neurons (including neurogliaform cells). We reasoned that the selective labeling could be due to (1) the transgenic lines used, (2) the serotype of the vector used, and/or (3) the location of the injections (especially as neurogliaform cells have dense, but radially very restricted, processes *Armstrong et al., 2012*). Crossing the nNOS-Cre and

Dlx5/6-Flpe lines with a DUAL reporter mouse (RCE:dual) resulted in an expression pattern that was consistent with the expression of nNOS in known interneuron populations (n = 1 Cre+/Flpe+/eGFP+ animal; *Figure 2—figure supplement 2*), suggesting that neither the Cre nor the Flpe line was responsible for the selective expression of ChR2-eYFP in LINCs. We next tested whether the DJ serotype used was critical. Injection of an AAV5 serotype of the Con/Fon-hChR2(H134R)-eYFP vector also resulted in the labeling of LINCs (n = 4 Cre+/Flpe+ animals; *Figure 2—figure supplement 2*), arguing against a strict requirement for the DJ serotype. Finally, we tested whether the location of the viral vector was critical. For labeling of LINCs, virus was targeted to the stratum oriens. If we instead injected AAV-DJ-hSyn-Con/Fon-hChR2(H134R)-eYFP-WPRE in the stratum lacunosum molecu-lare, cells with morphologies consistent with neurogliaform cells were indeed labeled (*Figure 2—figure supplement 2*). This illustrates that the intersectional approach taken is capable of labeling well-characterized populations of nNOS-expressing cells, but preferentially targets LINCs when injected into the stratum oriens.

Together, these findings suggested that this vector-based transgenic approach was not produc-ing inappropriate 'leaky' labeling, but was labeling a unique and unexpected population of nNOS+ neurons. This population of neurons had been mentioned only briefly in previous literature, and war-ranted further investigation.

## Morphological and electrophysiological properties of LINCs

As our findings suggested that LINCs had not been previously studied, we first sought to better characterize these cells by performing whole-cell patch clamp recordings from LINCs to determine their electrophysiological properties and provide greater examination of their morphologies.

LINCs had a modest input resistance (177 ± 17 MΩ (mean ± SEM); n = 21 cells from 21 sections, 13 animals), a threshold for firing near −44 mV (−44.6 ± 0.5 mV; n = 21 cells from 21 sections, 13 animals), and a relatively low firing frequency near threshold (31 ± 6 Hz; n = 21 cells from 21 sec-tions, 13 animals). LINCs also displayed subtle variability in their firing properties, which, to a limited extent, corresponded to dendritic morphology (*Figure 2b,c* insets; *Table 1*). Specifically, maximum firing frequency (hLINCs: 149 ± 22 Hz; vLINCs: 78 ± 18 Hz; hLINC vs vLINC, uncorrected p=0.02, two-tailed Mann-Whitney (M-W) test; n = 10 hLINCs from 10 sections, nine animals and n = 10 vLINCs from 10 sections, seven animals), adaptation ratio (hLINCs: 0.39 ± 0.05; vLINCs: 0.63 ± 0.05; hLINC vs vLINC uncorrected p=0.009, M-W; n = 10 hLINCs from 10 sections, nine animals and n = 10 vLINCs from 10 sections, seven animals) and coefficient of variance of the interspike interval (hLINCs: 11.7 ± 2.6; vLINCs: 24.4 ± 3.6; hLINC vs vLINC uncorrected p=0.009, M-W; n = 10 hLINCs from 10 sections, nine animals and n = 10 vLINCs from 10 sections, seven animals) were suggestive of differences, with hLINCs showing a slightly faster and more consistent rate of firing. However, no significant differences were noted in resting membrane potential (hLINCs: −55.1 ± 2.5 mV; vLINCs: −59.7 ± 3.1 mV; hLINC vs vLINC uncorrected p=0.35, M-W; n = 8 hLINCs from eight sections, seven animals and n = 10 vLINCs from 10 sections, seven animals), input resistance (hLINCs: 169 ± 20 MΩ; vLINCs: 191 ± 30 MΩ; hLINC vs vLINC uncorrected p=0.62, M-W; n = 10 hLINCs from 10 sections, nine animals and n = 10 vLINCs from 10 sections, seven animals), threshold voltage (hLINCs: −44.7 ± 0.6 mV; vLINCs: −44.2 ± 1.0 mV; hLINC vs vLINC uncorrected p=0.52, M-W; n = 10 hLINCs from 10 sections, nine animals and n = 10 vLINCs from 10 sections, seven animals), firing frequency near threshold (hLINCs: 33 ± 11 Hz; vLINCs: 30 ± 8 Hz; hLINC vs vLINC uncorrected p=0.85, M-W; n = 10 hLINCs from 10 sections, nine animals and n = 10 vLINCs from 10 sections, seven animals), spike amplitude ([i] at threshold — hLINCs: 54.4 ± 2.8 mV; vLINCs: 52.9 ± 1.5 mV; hLINC vs vLINC uncorrected p=0.62, M-W; [ii] at max firing — hLINCs: 39.7 ± 2.9 mV; vLINCs: 41.3 ± 3.0 mV; hLINC vs vLINC uncorrected p=0.43, M-W; n = 10 hLINCs from 10 sections, nine animals and n = 10 vLINCs from 10 sections, seven animals), action potential half width ([i] at threshold — hLINCs: 0.67 ± 0.10 ms; vLINCs: 0.96 ± 0.15 ms; hLINC vs vLINC uncorrected p=0.16, M-W; [ii] at max firing — hLINCs: 0.72 ± 0.15 ms; vLINCs: 1.20 ± 0.19 ms; hLINC vs vLINC uncorrected p=0.57, M-W; n = 10 hLINCs from 10 sections, nine animals and n = 10 vLINCs from 10 sections, seven animals), or proportion showing persistent firing (0 of 9 hLINCs (nine sections, eight animals), 2 of 10 vLINCs (10 sections, seven animals), uncorrected p=0.16, $\chi^2$ test). *Table 1* provides a detailed summary of the electrophysiological properties of LINCs by dendritic morphology. Overall, LINCs had firing properties (*Figure 2b and c* insets) consistent with 'regular spiking' or 'regular spiking non-pyrami-dal' descriptions (*Kawaguchi and Kubota, 1998*; *Krook-Magnuson et al., 2008*).

**Table 1.** Electrophysiological properties of LINCs by dendritic morphology.

| | Horizontal mean ± SEM (median) | Vertical mean ± SEM (median) | p-values (corrected α = 0.0045) |
|---|---|---|---|
| Vrest (mV) | −55.1 ± 2.5 (−52.8) | −59.7 ± 3.1 (−56.9) | 0.35 |
| Input resistance (MΩ) | 169 ± 20 (181) | 191 ± 30 (198) | 0.62 |
| Threshold voltage (mV) | −44.7 ± 0.6 (−45.0) | −44.2 ± 1.0 (−44.0) | 0.52 |
| Firing frequency near threshold (Hz) | 33 ± 11 (13) | 30 ± 8 (18) | 0.85 |
| Max firing frequency (Hz) | 149 ± 22 (160) | 78 ± 18 (55) | 0.02 |
| Adaptation ratio of the interspike interval (ISI) at max firing | 0.39 ± 0.05 (0.41) | 0.63 ± 0.05 (0.67) | 0.009 |
| Spike amplitude at threshold (mV) | 54.4 ± 2.8 (54.8) | 52.9 ± 1.5 (52.6) | 0.62 |
| Spike amplitude at max firing frequency (mV) | 39.7 ± 2.9 (38.6) | 41.3 ± 3.0 (43.6) | 0.43 |
| Action potential half width at threshold (ms) | 0.67 ± 0.10 (0.55) | 0.96 ± 0.15 (0.92) | 0.16 |
| Action potential half width at max firing frequency (ms) | 0.72 ± 0.15 (0.83) | 1.20 ± 0.19 (1.17) | 0.57 |
| Coefficient of variance of the ISI at max firing | 11.7 ± 2.6 (10.2) | 24.4 ± 3.6 (23.3) | 0.009 |
| Proportion of cells showing persistent firing | 0/9 | 2/10 | 0.16 |

Two-tailed Mann-Whitney tests performed with Bonferroni corrected α; $\chi^2$ test performed for comparison of proportions showing persistent firing. Note that persistent firing (**Krook-Magnuson et al., 2011**; **Sheffield et al., 2011**), also known as axonal barrage firing (**Sheffield et al., 2013**), is associated with a different population of nNOS-expressing CA1 interneurons (i.e., 80% of neurogliaform cells display persistent firing, whereas only ~20% of cells of other interneuron types display this phenomenon [**Krook-Magnuson et al., 2011**; **Sheffield et al., 2013**; **Suzuki et al., 2014**]), but is only rarely found in LINCs. Note also that despite potential subtle differences in firing pattern, LINCs show similar thresholds, input resistance, and firing frequency near threshold regardless of their dendritic morphology. Individual data points for each cell are included in **Table 1—source data 1.**

DOI: https://doi.org/10.7554/eLife.46816.008

The following source data is available for Table 1:

Source data 1. Source data for the electrophysiological properties of individual LINCs.

Data collected from whole cell patch clamp recordings from individual LINCs in CA1, including neuroanatomical descriptions such as dendritic morphology and cell body location, and section orientation. These data are summarized across horizontal and vertical LINCs in **Table 1**.

DOI: https://doi.org/10.7554/eLife.46816.009

Inhibitory neurons are typically categorized primarily by their axonal, rather than dendritic, arbor (**Freund and Buzsáki, 1996**; **Klausberger and Somogyi, 2008**). Regardless of slice orientation, many LINCs displayed poor axonal recovery, with the axon often lost as it ascended towards the alveus, suggestive long-range projections (**Gulyás et al., 2003**; **Villette et al., 2016**). Indeed, all filled LINCs displayed an axon that was lost as it exited the sectioned tissue (21/21 filled LINCs from 21 sections, 13 animals; **Figure 2b,c**, **Figure 2c** inset box, **Figure 2—figure supplement 3**). For the cells in which a more extensive local axonal arbor was recovered, a variety of axonal morphologies were found (**Figure 2b,c**, **Figure 2—figure supplement 3**), which did not appear to correspond strongly to dendritic morphology (**Figure 2b,c**, **Figure 2—figure supplement 3**). No drumstick-like appendages (**Gulyás et al., 2003**; **Sik et al., 1994**) were noted on the axons of LINCs. Despite the generally poor axonal recovery from LINCs filled during ex vivo hippocampal recordings, we found that, collectively, local axons reached all layers of CA1 (**Figure 1d**, **Figure 2b,c**, **Figure 2—figure supplement 3**), suggesting a potentially broad impact of LINCs on neurons in the region. We therefore asked next what populations of CA1 cells are targeted by these local axons.

## LINCs provide broad and long-lasting inhibition to CA1

Previous work indicates heterogeneity in hippocampal pyramidal cells (*Graves et al., 2012*; *Hunt et al., 2018*; *Mizuseki et al., 2011*; *Valero and de la Prida, 2018*), including heterogeneity in their inhibition by local interneurons (*Lee et al., 2014*; *Valero et al., 2015*). Therefore, to determine LINCs' local connectivity, we recorded from both deep and superficial pyramidal cells, as well as from inhibitory neurons across all layers of CA1, while optogenetically activating LINCs. We found that LINCs broadly targeted both deep and superficial CA1 pyramidal cells (dPC and sPC, respectively; *Figure 3*), and to a roughly equivalent degree: GABA$_A$ responses (subsequently blocked by 5 μM gabazine) were recorded in approximately 80% of dPCs and sPCs (13/16 dPCs; 16/20 sPCs; p=0.93, $\chi^2$ test; n = 16 dPCs from 16 sections, 12 animals, n = 20 sPCs from 19 sections, 11 animals; *Figure 3a,d*), and were of similar amplitude in both dPCs and sPCs ([i] GABA$_A$ amplitude median dPC: −106 pA; sPC: −83 pA; dPCs vs sPCs, p=0.97, M-W; n = 13 dPCs from 13 sections, 10 animals, and n = 16 sPCs from 16 sections, nine animals; *Figure 3b*; [ii] including non-responders, dPC: −87 pA; sPC: −66 pA, dPCs vs sPCs, p=0.29, M-W, n = 16 dPCs from 16 sections, 12 animals, and n = 20 sPCs from 19 sections, 11 animals). The rise time of the GABA$_A$ response (time between 0% and 63% of peak response) was also similar in both dPCs and sPCs (rise time dPC: 3.26 ± 1.00 ms; sPC: 3.64 ± 0.77 ms; dPCs vs sPCs, p=0.78, M-W; n = 13 dPCs from 13 sections, 10 animals, and n = 16 sPCs from 16 sections, nine animals). This suggests that LINCs provide similarly broad inhibition to both deep and superficial pyramidal cells in CA1, and therefore could link these two distinct information processing streams (*Valero and de la Prida, 2018*; *Krook-Magnuson et al., 2012*). Notably, LINCs also displayed similarly broad targeting of inhibitory neurons (INs), with approximately 80% of recorded INs also showing a postsynaptic GABA$_A$ response (26/34 INs; vs dPCs p=0.70 $\chi^2$; vs sPCs p=0.76, $\chi^2$; [i] median: −114 pA; INs vs dPC vs sPCs, p=0.88, Kruskal-Wallis ANOVA (K-W), n = 26 INs from 26 sections, 14 animals; [ii] median including non-responders: −80 pA; IN vs dPC vs sPC, p=0.62, K-W; n = 34 INs from 34 sections, 16 animals; *Figure 3*), with slightly stronger inhibition provided to INs with somata in the stratum pyramidale (*Figure 3—figure supplement 1*). Taken together, these findings indicate that LINCs provide unusually broad inhibition to CA1 cells.

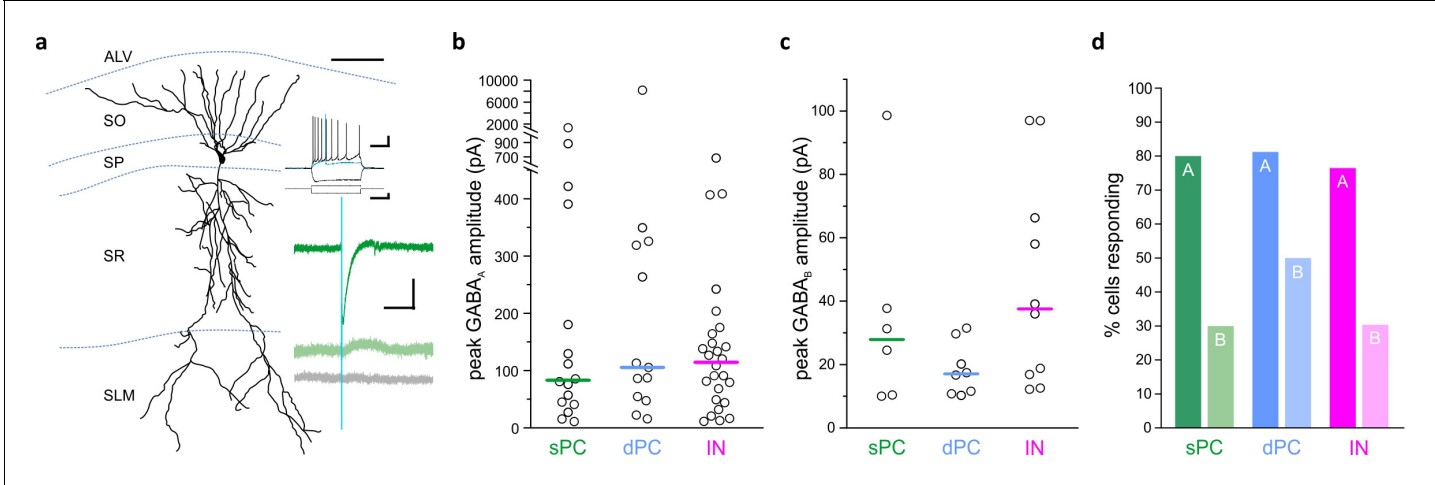

**Figure 3.** LINCs provide broad and long-lasting inhibition. Optogenetic activation of LINCs produces postsynaptic inhibitory responses in superficial pyramidal cells (sPC, green; example morphology, firing properties, and light-evoked inhibitory post-synaptic currents (IPSCs) are shown in panel [a]), deep pyramidal cells (dPC, blue), and inhibitory neurons (IN, pink). (a) Top green trace, light-evoked IPSC; middle green trace, in gabazine; bottom gray trace, in gabazine plus CGP55845. (b, c) Peak amplitudes of GABA$_A$ (b) or GABA$_B$ responses (c) in individual cells showing a response. Bar denotes median amplitude. (d) Percentage of sPCs, dPCs, and INs with GABA$_A$ response (denoted with 'A') or a GABA$_B$ response ('B'). Scale bars (a): 50 μm; 20 mv, 200 ms (top set); 400 pA, 200 ms (middle); 50 pA, 200 ms (bottom).

DOI: https://doi.org/10.7554/eLife.46816.010

The following figure supplement is available for figure 3:

**Figure supplement 1.** LINCs produce fast and long-lasting inhibitory postsynaptic currents in a variety of inhibitory neurons in CA1.
DOI: https://doi.org/10.7554/eLife.46816.011

In addition to a postsynaptic GABA$_A$ response, a remarkably large percentage of recorded cells also displayed a postsynaptic GABA$_B$-mediated response (present in gabazine, but blocked by the subsequent application of 5 μM CGP55845; note that GABA$_B$ responses were only recorded in cells that also showed a GABA$_A$ response) (*Figure 3*). No postsynaptic responses remained after the application of both GABA receptor antagonists (*Figure 3a*), further confirming the specificity of ChR2 targeting to only GABAergic neurons. Approximately 30–50% of recorded cells displayed a postsynaptic GABA$_B$-mediated response (8/16 dPCs, 6/20 sPCs, and 10/34 INs; dPCs vs sPCs p=0.22, dPCs vs INs p=0.16, sPCs vs INs p=0.96, $\chi^2$; *Figure 3d*), with roughly equal amplitude across cell groups ([i] median —dPC: 17 ± 3 pA; sPC: 28 ± 13 pA; IN: 38 ± 10 pA; dPC vs sPC vs INs p=0.14, K-W; n = 8 dPCs from eight sections, seven animals, n = 6 sPCs from six sections, five animals, and n = 10 INs from 10 sections, six animals; [ii] median including non-responders — dPCs: 7 ± 3 pA; sPCs: 3 ± 6 pA; INs: 4 ± 5 pA; p=0.28, K-W; n = 16 dPCs from 16 sections, 12 animals, n = 20 sPCs from 19 sections, 11 animals, and n = 34 INs from 34 sections, 16 animals; *Figure 3c*). Taken together, these data indicate that LINCS provide strong, broad inhibition (through GABA$_A$-mediated inhibition) as well as relatively long-lasting (GABA$_B$-mediated) inhibition in CA1. This would place LINCs in an influential position, capable of having a major impact on hippocampal function.

## LINCs have long-range projections

While the majority of hippocampal GABAergic neurons are true interneurons, with axons limited to targeting local neurons, there are notable exceptions, including cell-types that project far outside the hippocampus (*Jinno et al., 2007*; *Katona et al., 2017*; *Melzer et al., 2012*). As noted above, recovered morphologies of LINCs that were recorded ex vivo were suggestive of long-range, extra-hippocampal projections. We further examined this possibility in X-CLARITY cleared tissue, as well as in traditionally sectioned tissue (*Figure 4*, *Figure 4—figure supplement 1*). Fibers exiting the hippocampus through the fimbria were clearly visible (*Figure 4a*, *Figure 4—video 1*). These fibers continued down through the medial septum, and into the dorsal and ventral tenia tecta and the vertical and horizontal limbs of the diagonal band of Broca (*Figure 4b,c*). Quantification of eYFP fibers in the septum and diagonal band indicated that LINCs have dense projections to these areas, although these are not quite as dense as somatostatin-expressing projections (e.g. hippocampal-septal) cells (*Figure 4—figure supplement 2*). In the medial septum, the average fiber length of LINCs (per fiber) was actually longer than that of SOM+ projections (164 μm vs 95 μm, respectively) because LINCs had significantly fewer short fibers in the medial septum (distribution of LINC fiber length in the medial septum vs. hippocampal SOM+ fiber length in the medial septum, p<0.01, Kolmogorov-Smirnov (K-S) test, data from n = three 50 μm sections from three animals, each; *Figure 4—figure supplement 2*). This suggests that LINCs send projections through the medial septum (with limited local branching), whereas SOM+ hippocampal-septal cells have more branches within the medial septum.

Quantification of eYFP fibers further indicated that LINCs uniquely provide strong input to the tenia tecta, as hippocampal SOM+ cells had virtually no fibers there (*Figure 4—figure supplement 2*). LINCs also have broad connections to several other extrahippocampal regions: LINC projections were also identified in the dorsal subiculum, entorhinal cortex, mammillary nuclei, lateral hypothalamus, olfactory tubercle, olfactory bulb, ipsilateral dentate gyrus (with some fibers projecting through CA3 and others crossing the hippocampal fissure), and the contralateral hippocampal formation (including the contralateral dentate gyrus and fibers visible in the dorsal hippocampal commissure) (*Figure 4—figure supplement 1*). To further confirm that CA1 LINCs project directly to extrahippocampal areas, we injected the retrograde tracer Fluorogold (FG) into the tenia tecta, medial septum, or diagonal band of Broca (*Figure 4e*; *Figure 4—figure supplement 2*). As expected, co-labeling with eYFP and FG in CA1 LINCs was observed in all instances, further supporting the observation that eYFP+ fibers in target areas arise from LINCs, and are not due to unobserved leaky expression in pyramidal cells nor to any weak expression along the injection tract in the overlying cortex.

In addition to extensive targeting within CA1, LINCs provide long-range input to a variety of extrahippocampal brain regions, positioning them to play a role in inter-regional communication or synchrony.

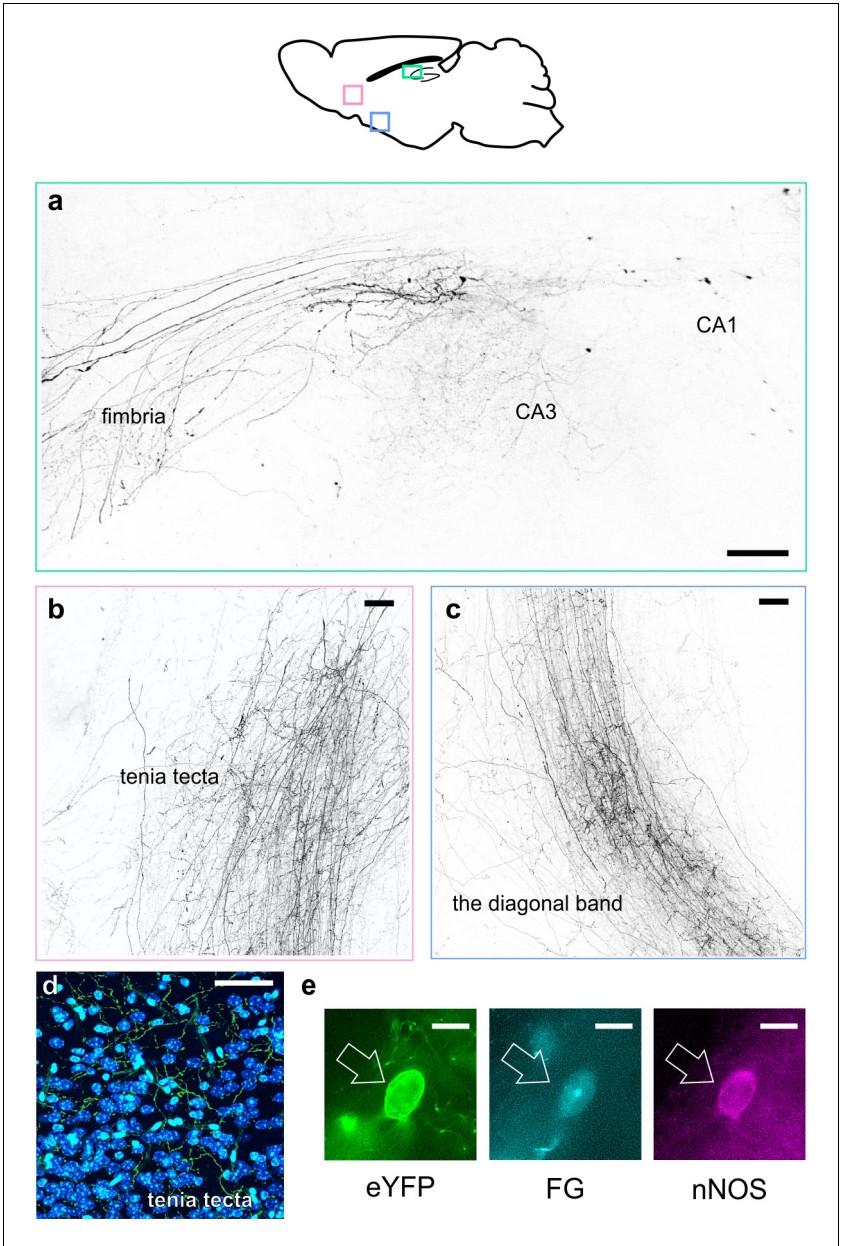

**Figure 4.** LINCs have extrahippocampal projections. LINCs project out of the hippocampus (**a**), through the medial septum, and into the tenia tecta (**b, d**), the diagonal band (**c**), and other areas (*Figure 4—figure supplement 1*). (**a–c**) Max projections from X-CLARITY cleared tissue. (**d**) eYFP+ processes viewed in the dorsal tenia tecta from a 50 µm section; DAPI in blue. (**e**) Example LINC colabeled with the retrograde tracer Fluorogold (FG) and confirmed immunoreactive for nNOS following injection of FG into the diagonal band. Scale bars: 100 µm (**a–c**), 50 µm (**d**), and 10 µm (**e**).

DOI: https://doi.org/10.7554/eLife.46816.012

The following video and figure supplements are available for figure 4:

**Figure supplement 1.** LINCs have long-range projections to several extrahippocampal regions.
DOI: https://doi.org/10.7554/eLife.46816.013

**Figure supplement 2.** Quantification of LINC fibers.
DOI: https://doi.org/10.7554/eLife.46816.014

**Figure 4—video 1.** eYFP+ fibers exiting the hippocampus through the fimbria.
DOI: https://doi.org/10.7554/eLife.46816.015

## Immunohistochemical profile of LINCs

To characterize CA1 LINCs more fully and to allow comparison to previously described hippocampal GABAergic populations, we performed immunohistochemistry for neuropeptide Y (NPY), somatostatin (SOM), parvalbumin (PV), vasoactive intestinal peptide (VIP), calretinin (CR), and the muscarinic acetylcholine receptor M2 (M2R). Although a substantial number of LINCs were NPY immunopositive (NPY+; 46.9 ± 9.7%, n = 3 animals, mean ± SD; *Figure 5*), only a small subset of LINCs were immunopositive for SOM (SOM+; 4.9 ± 4.4%, n = 3 animals; *Figure 5*; *Figure 5—figure supplement 1*), and even fewer LINCs were immunopositive for both SOM and NPY (1.8 ± 1.6%, n = 3 animals). This immunohistochemical profile is inconsistent with LINCs simply being a hippocampally located version of neocortical NOS-type I cells, or previously described hippocampal-septal cells, double-projection cells, or back-projection cells (*Jinno et al., 2007*; *Klausberger, 2009*). However, our data suggest that the ~5% of LINCs that are SOM+ represent partial overlap between LINCs and SOM+ hippocampal-septal cells (*Figure 4—figure supplement 2*).

Other hippocampal GABAergic projection neurons, including trilaminar cells and oriens-retrohippocampal cells, are commonly associated with the expression of M2R (*Jinno et al., 2007*; *Klausberger, 2009*). We therefore also tested LINCs for this marker, but again found relatively few LINCs that were immunolabeled (M2R-immunopositive CA1 LINCs: 8.5 ± 3.2%, n = 3 animals; *Figure 5*; *Figure 5—figure supplement 1*). Previous work has shown that hippocampal PV+ cells can also have long-range projections, targeting the contralateral hippocampus (*Christenson Wick et al., 2017*; *Eyre and Bartos, 2019*; *Goodman and Sloviter, 1992*). However, we found only limited co-labeling of LINCs with PV (PV-immunopositive CA1 LINCs: 20.9 ± 7.7%, n = 3 animals; *Figure 5*; *Figure 5—figure supplement 1*).

Recent reports have also described VIP-expressing projection neurons that target the subiculum (*Francavilla et al., 2018*; *Luo et al., 2019*). We therefore also examined LINCs for VIP-immunoreactivity, but found that very few LINCs were VIP immunopositive (8.8 ± 2.3%, n = 3 animals). The presence of so few LINCs expressing VIP also suggests that LINCs' expression profiles do not correspond with those previously described for nNOS-expressing type III interneuron-selective-interneurons, which express VIP and CR (*Tricoire and Vitalis, 2012*; *Tricoire et al., 2010*). Nevertheless, to explore this further, we also performed immunohistochemistry against CR. Only 4.0 ± 3.1% of LINCs colocalized with CR (n = 3 animals), and even fewer were immunopositive for both VIP and CR (3.3 ± 2.0%, n = 3 animals; *Figure 5*).

Taken together, it is evident that LINCs are best labeled with nNOS (although nNOS expression is not limited to LINCs [*Armstrong et al., 2012*; *Freund and Buzsáki, 1996*; *Tricoire and Vitalis, 2012*]), rather than with other canonical markers of hippocampal inhibitory projection neurons, and that LINCs do not fit well into previously described CA1 GABAergic cell populations.

## LINCs are generated around E11

In order to assess the likely birthdate of LINCs, we injected pregnant dams with BrdU at embryonic days (E) 9.5, E10.5, E11.5, or E13.5, and then subsequently injected the adult offspring with AAV-DJ-hSyn-Con/Fon-hChR2(H134R)-eYFP-WPRE. Six weeks after viral injection, tissue was then processed to determine the degree of colocalization of BrdU with eYFP. Virtually no LINCs were BrdU labeled following BrdU injections at E9.5 or E13.5 (E9.5: 0.0 ± 0.0% CA1 LINCs colocalized with BrdU, n = 2 animals, one litter; E13.5: 0.3 ± 0.5%, n = 3 animals, one litter, mean ± SD). By contrast, eYFP and BrdU were found to colocalize following BrdU injections at E10.5 or E11.5 (E10.5: 6.8 ± 1.9%, n = 3 animals, one litter; E11.5: 16.0 ± 3.1%, n = 3 animals, one litter, mean ± SD; *Figure 6*). This places the birthdate of LINCs after the reported birthdates of early-generated GABAergic 'hub' neurons (*Picardo et al., 2011*; *Villette et al., 2016*).

## LINCs impact hippocampal function

Despite being relatively sparse in number (on average, approximately 244 eYFP+ cells were counted after dorsal virus injection, counting cells in every fourth 50 µm section, n = 9 animals), LINCs have long-range projections, which suggest a role in inter-regional communication, and provide broad local inhibition, suggesting an influential role in the hippocampus. We therefore next set out to test how the optogenetic activation of LINCs might influence hippocampal function in vivo. To test this, we optogenetically manipulated LINCs in vivo during the object location memory (OLM) task and

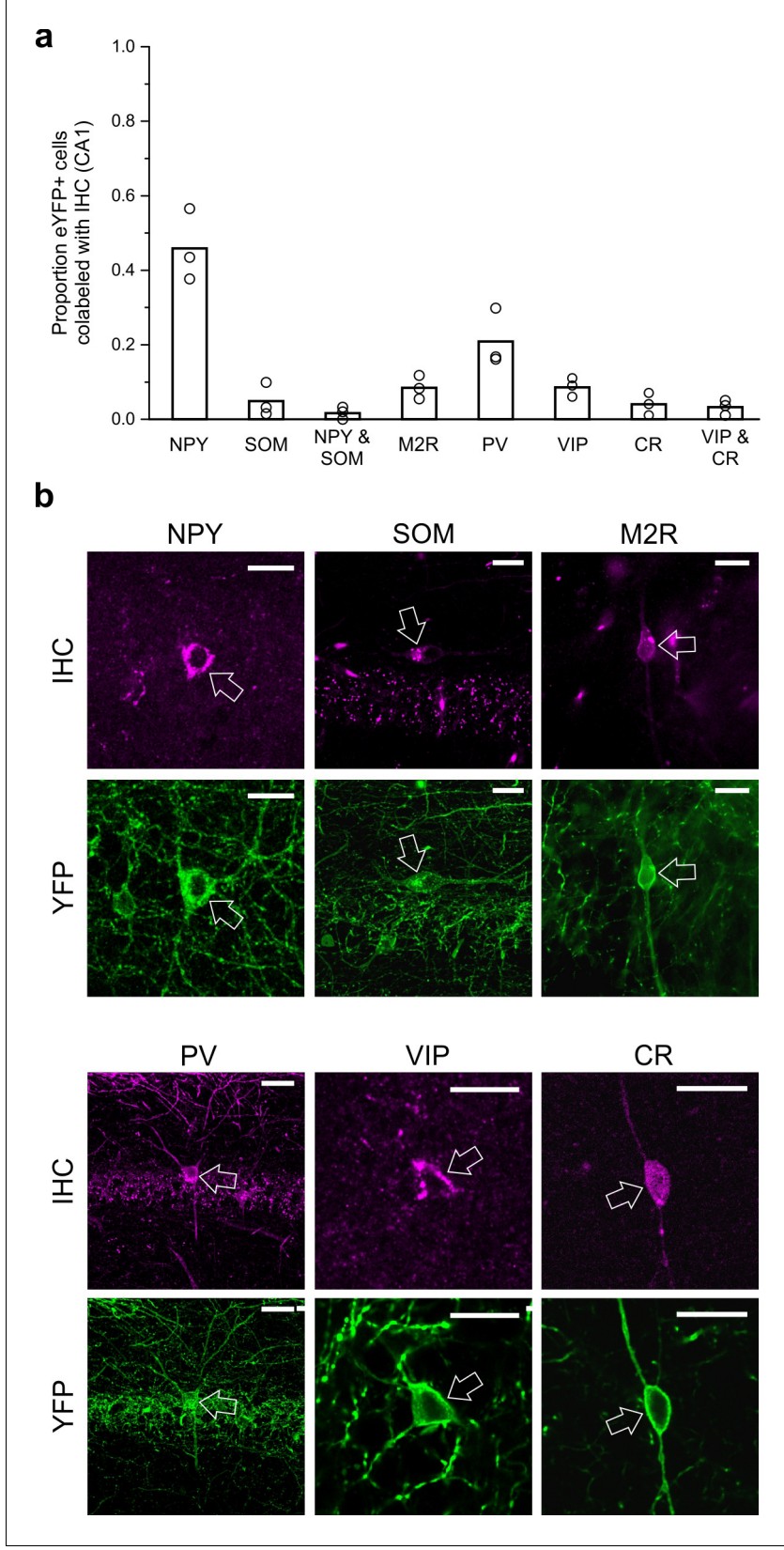

**Figure 5.** LINCs can express NPY but typically do not express SOM, M2R, PV, VIP, or CR. Virally labeled (eYFP+) somata in CA1 were examined for immunofluorescent colocalization with various molecular markers or combinations of markers associated with other long-range projecting inhibitory cells or other nNOS+ interneurons. *Figure 5 continued on next page*

*Figure 5 continued*

(a) Although a substantial number of eYFP+ cells were immunopositive for neuropeptide Y (NPY), few were immunopositive for somatostatin (SOM), and even fewer for both NPY and SOM, or for muscarinic acetylcholine receptor M2 (M2R), parvalbumin (PV), vasoactive intestinal peptide (VIP), and/or calretinin (CR). (b) Representative images of viral colocalization with immunofluorescence (arrows). Data in panel (a) are means, circles represent data from individual animals. Virally labeled cells were counted and viewed for immunofluorescence in every fourth 50 µm hippocampal section in n = 3 animals following a viral injection targeting dorsal CA1. An additional three animals were instead injected with virus targeting towards ventral CA1 and assessed for colocalization with SOM, M2R, and PV (data shown in *Figure 5—figure supplement 1*) and show largely the same pattern of molecular expression. Scale bars = 10 µm.

DOI: https://doi.org/10.7554/eLife.46816.016

The following figure supplement is available for figure 5:

**Figure supplement 1.** Expanded immunohistochemical data, including data for regions outside of CA1 following viral injection targeted to the dorsal or ventral hippocampus.

DOI: https://doi.org/10.7554/eLife.46816.017

the object recognition memory (ORM) task (*Figure 7a*). The ORM and the OLM tasks are nicely parallel in format, but the OLM task is strongly hippocampal-dependent while the ORM task is not (*Barker and Warburton, 2011*; *Leger et al., 2013*; *McNulty et al., 2012*; *Stefanko et al., 2009*).

Optogenetic manipulation of LINCs at ~7 Hz (50 ms pulse width, 33% duty cycle, as in *Krook-Magnuson et al., 2013*) for 3 s every 30 s during encoding and retrieval had no significant effect on performance of the ORM task (ORM discrimination index: 19.9 ± 4.5 [mean ± SEM] [n = 16] for opsin-positive animals vs 31.7 ± 7.3 [n = 8] for opsin-negative animals, p=0.34, M-W, mean ± SEM; *Figure 7b*), but produced strong spatial memory deficits in the OLM task (OLM discrimination index: 10.1 ± 4.7 [mean ± SEM] [n = 14] for opsin+ animals vs 31.4 ± 5.2 [n = 8] for opsin-negative animals; p=0.009, M-W, ; *Figure 7b*). Confirming that there was no interference with object exploration itself (i.e. that effects on the discrimination index were neither due to a general indifference to the objects nor to motor deficits), there was no significant difference in the total time spent exploring objects during either task (OLM time spent exploring: 9.5 ± 1.0 s for opsin+ animals vs. 7.0 ± 1.3 s for opsin– animals; p=0.16; ORM time spent exploring: 12.9 ± 1.5 s for ORM opsin+ animals vs. 16.0 ± 4.7 s for opsin– animals; p=0.98; *Figure 7c*). These findings illustrate that despite being relatively few in

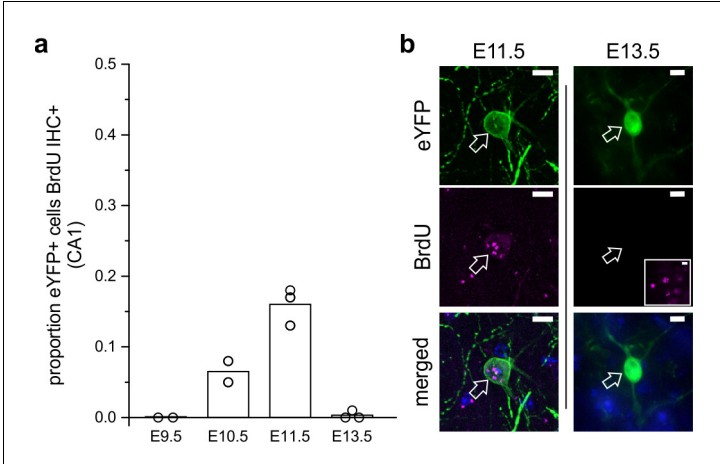

**Figure 6.** LINCs are born on or around embryonic day 11. (a) The proportion of eYFP+ cells in CA1 that colocalized with BrdU in adult mice that were previously exposed to BrdU at embryonic day 9.5 (E9.5), 10.5, 11.5, or 13.5. (b) Representative images of eYFP colocalizing with BrdU at E11.5 but not at E13.5. The inset shows BrdU elsewhere in the hippocampus at E13.5 with the same image acquisition settings. Data are shown as means; circles represent data from individual animals. Scale bars = 10 µm.

DOI: https://doi.org/10.7554/eLife.46816.018

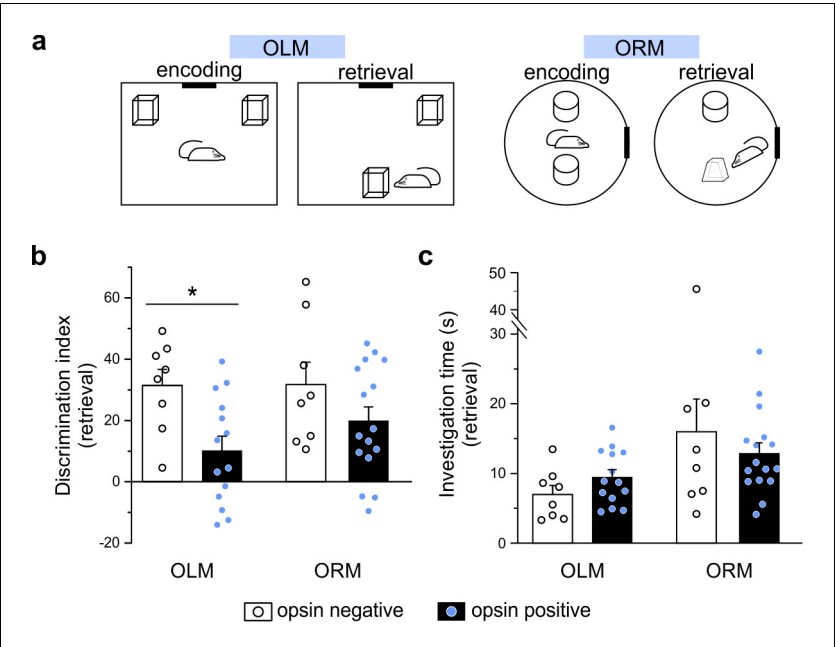

**Figure 7.** Optogenetic manipulation of LINCs in vivo affects spatial memory. (**a**) Schematics for object location (OLM) and recognition memory (ORM) tasks. (**b**) Light stimulation significantly decreases performance (i.e. discrimination index) in opsin-positive animals in the OLM task but not in the ORM task. (**c**) Light stimulation does not affect total object investigation time during retrieval. (**b, c**) Data shown: mean ± SEM, with circles representing data points from individual animals. Asterisk: p-value<0.05, M-W, opsin+ vs opsin-negative.
DOI: https://doi.org/10.7554/eLife.46816.019

The following figure supplement is available for figure 7:

**Figure supplement 1.** Manipulating LINCs did not impact performance in an odor recognition memory task.
DOI: https://doi.org/10.7554/eLife.46816.020

number, LINCs are able to impact hippocampal function, and that artificially manipulating LINC activity can have a detrimental impact on spatial/location memory.

To further examine this effect, we recorded hippocampal local field potentials (LFPs) while optogenetically stimulating LINCs with the same pulsed light frequency used during the OLM and ORM tasks (~7 Hz stimulation, 33% duty cycle, delivered for 3 s every 30 s). First, we examined all traces and found an increase in power at the stimulation frequency in the hippocampal LFP in opsin expressing animals (62.3 ± 17.9% [mean ± SEM] [n = 12] for opsin+ animals vs. −11.9 ± 10.3% [n = 6] for opsin– animals; p=0.006, M-W; *Figure 8*). Given that the animal would already be in a theta state during most of the OLM or ORM task, we next specifically looked at traces recorded when the animal was in a theta state at the time of light delivery. Under these circumstances, we found an increase in overall theta power during light stimulation in opsin-positive animals compared to opsin-negatives (43.6 ± 28.7% increase [n = 12] for opsin+ animals vs. −11.0 ± 7.2% [n = 6]for opsin– animals; p=0.006, M-W). Moreover, we found that the theta oscillations had reset to align to the light delivery in opsin-positive (*Figure 8g*) but not opsin-negative (*Figure 8h*) animals. This entrainment to the light also resulted in a shift in the dominant frequency within theta towards the ~7 Hz light-delivery frequency in opsin-positive animals (7.6 ± 0.2 Hz prior to light stimulation vs. 6.9 ± 0.1 Hz during stimulation; p=0.002, M-W; *Figure 8c*). To further examine the impact of this light delivery scheme on both LINCs and their downstream targets, we returned to slice electrophysiology. Light delivery at ~7 Hz with 50 ms pulse width for 3 s produced action potentials in LINCs with each light pulse, and corresponding IPSCs in postsynaptic neurons (*Figure 8—figure supplement 1*). In summary, optogenetic activation of LINCs within theta ranges (~7 Hz) causes inhibition in post-synaptic neurons, an entrainment of the theta LFP to the light, and impairment of hippocampal function.

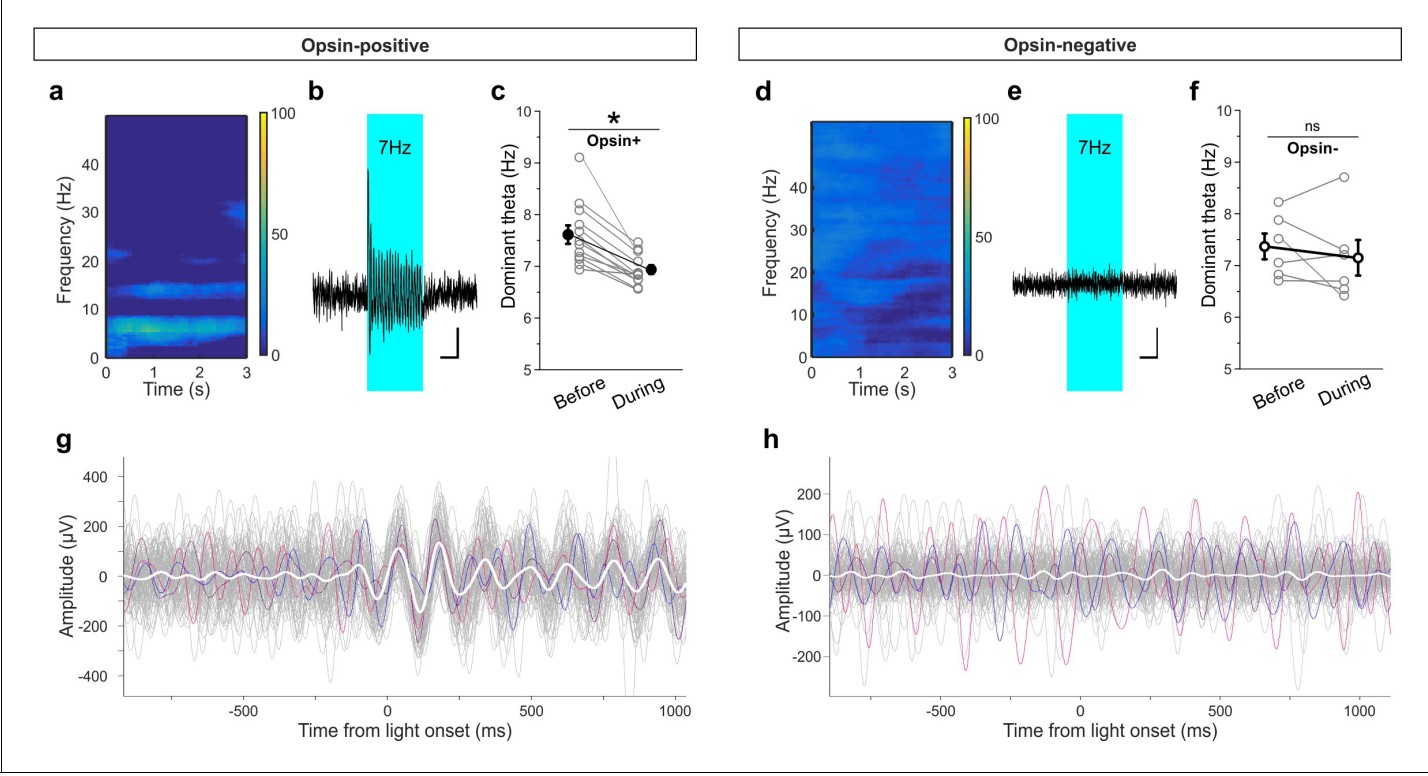

**Figure 8.** Light delivery alters hippocampal theta oscillations. Light delivery at ~7 Hz (50 ms on, 100 ms off) for 3 s once every 30 s increases hippocampal power at and around 7 Hz in opsin-positive animals (spectrogram shows averaged percent change in hippocampal power during a period of light delivery from n = 12 opsin+ animals) (a) but not opsin-negative (d) animals (n = 6 opsin– animals). (b, e) Average unfiltered traces showing LFP response to the pulsed light in one opsin+ (b) and one opsin– (e) animal. The blue boxes indicate the period of light delivery. (c, f) Dominant frequency within the theta range is pulled towards the light delivery frequency in opsin+ (c) but not opsin– (f) animals. Gray circles show individual animals' average dominant theta frequency during trials when the animals were in native theta states; bold circles show means across animals; error bars represent SEM. (g, h) Light delivery at ~7 Hz disrupts on-going theta oscillations (band pass filtered between 5–12 Hz) by inducing an alignment to light delivery. Example traces (gray, colored) during theta states from one example opsin+ (g) or opsin– (h) animal. White shows the averaged trace. The asterisk in panel (c) indicates p<0.01, M-W test. Scale bars (b, e) = 1 s, 25µV.

DOI: https://doi.org/10.7554/eLife.46816.021

The following figure supplement is available for figure 8:

**Figure supplement 1.** ~7Hz light stimulation produces LINC firing and IPSCs in their postsynaptic targets.
DOI: https://doi.org/10.7554/eLife.46816.022

## LINCs impact oscillations and HI-frontal cortex coherence across a range of frequencies

We next sought to determine the impact of optogenetic activation of LINCs on hippocampal and extrahippocampal network synchrony across a range of stimulation frequencies. To do this, we simultaneously monitored the hippocampal and the tenia tecta (TT; a frontal cortex brain region receiving input from CA1 LINCs; *Figure 4*) LFP, while optogenetically stimulating LINCs in the hippocampus.

Stimulating LINCs in the hippocampus, at a variety of frequencies (5 ms pulses), strongly increased hippocampal power at that stimulation frequency (mixed-design ANOVA with Greenhouse-Geisser correction: genotype p=0.00009, F = 27, degrees of freedom (DF) = 1; stimulation frequency*genotype p=0.05; F = 2.6, DF = 3.5; n = 12 opsin+ animals, n = 6 opsin– animals; *Figure 9c*; *Figure 9—figure supplements 1–4*), with stimulation between roughly 12 Hz and 30 Hz showing the greatest entrainment of the LFP to the stimulation frequency. We further examined the ability of LINCs to follow various light stimulation frequencies in hippocampal slices (which is determined by a number of interacting factors, including the kinetics of ChR2, expression levels of ChR2, and intrinsic cell properties). Consistent with this complex interaction of factors, in response to 5 ms

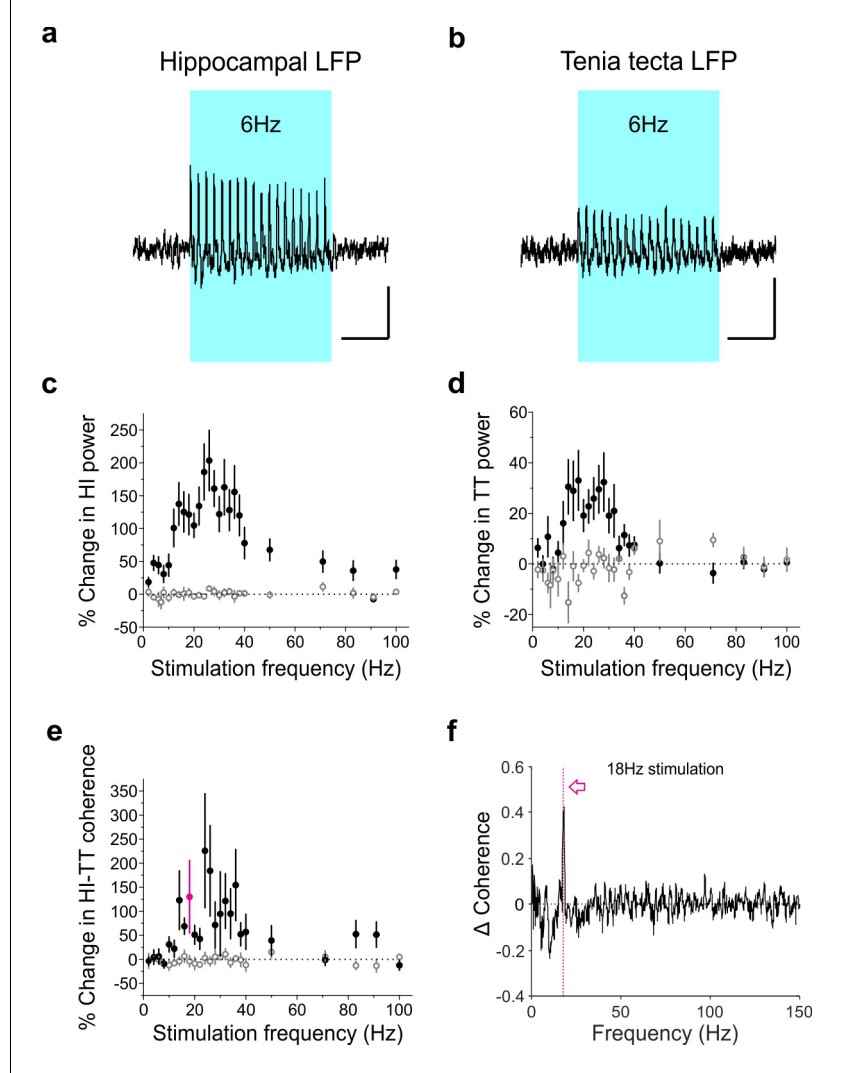

**Figure 9.** Optogenetic activation of LINCs produces a corresponding increase in hippocampal and tenia tecta power and coherence across a range of frequencies. (a–b) Averaged hippocampal (a) and tenia tecta (b) LFP traces from one opsin+ animal receiving 6 Hz blue light stimulation (5 ms pulse duration). (c–d) Percent change in hippocampal (HI) (c) and tenia tecta (TT) (d) power at the stimulation frequency during light stimulation (opsin+: closed circles; opsin–: open circles). (e) Percent change in hippocampal-tenia tecta coherence at the stimulation frequency. (f) Change in coherence in an opsin+ animal with 18 Hz stimulation (magenta). Data for individual animals are shown in *Figure 9—figure supplement 1* and average differential spectrograms for each stimulation frequency below 50 Hz are provided in *Figure 9—figure supplements 2–4*. Scale bars (a,b) = 1 s, 25 µV.

DOI: https://doi.org/10.7554/eLife.46816.023

The following figure supplements are available for figure 9:

**Figure supplement 1.** Expansion of in vivo electrophysiology, showing individual data points.
DOI: https://doi.org/10.7554/eLife.46816.024

**Figure supplement 2.** Differential spectrograms showing increases in hippocampal and TT power with different stimulation frequencies (Part 1).
DOI: https://doi.org/10.7554/eLife.46816.025

**Figure supplement 3.** Differential spectrograms showing increases in hippocampal and TT power with different stimulation frequencies (Part 2).
DOI: https://doi.org/10.7554/eLife.46816.026

**Figure supplement 4.** Differential spectrograms showing increases in hippocampal and TT power with different stimulation frequencies (Part 3).
DOI: https://doi.org/10.7554/eLife.46816.027

*Figure 9 continued on next page*

*Figure 9 continued*

**Figure supplement 5.** ChR2-expressing LINCs differ in their ability to follow light trains.
DOI: https://doi.org/10.7554/eLife.46816.028

blue light pulses, some LINCs fired action potentials for only a brief period, whereas others showed more prolonged periods of action potential firing, which could substantially outlast the duration of each light pulse (*Figure 9—figure supplement 5*). The duration of firing of action potentials per light pulse in turn determined the light stimulation frequency to which an individual LINC was able to entrain, and ranged from under 20 Hz to over 120 Hz for the LINCs sampled (*Figure 9—figure supplement 5*). This, together with considerations such as IPSC decay kinetics and network dynamics, are likely to shape which LFP oscillatory frequencies were most robustly altered by light stimulation, especially at higher frequencies.

As LINCs project outside of the hippocampus to areas including the TT, we were interested to see whether stimulation of LINCs was also able to induce changes in the TT LFP. Indeed, an increase in power at the stimulation frequency when optogenetically activating LINCs in the hippocampus was also apparent in the TT (mixed-design ANOVA with Greenhouse-Geisser correction: genotype p=0.018, F = 7, DF = 1; stimulation frequency*genotype p=0.01; F = 2.9, DF = 6.1, n = 11 opsin+ animals, n = 6 opsin− animals; *Figure 9d*; *Figure 9—figure supplements 1–4*), although this increase was smaller than that seen in the hippocampus itself (opsin+ animals, two-way repeated measures ANOVA with Greenhouse-Geisser correction: location p=0.00015, F = 35, DF = 1; location*frequency p=0.016, F = 3.7, DF = 3.6).

Beyond increases in oscillation power, increases in oscillation synchrony (i.e. coherence) are believed to play an important role in coordinating activity between brain regions (*Buzsáki and Schomburg, 2015*; *Colgin, 2011*; *Sirota et al., 2008*). LINCs, having connectivity both within the hippocampus and to extrahippocampal regions, are in a prime position to increase interregional coherence (i.e., to 'link' these regions up). Supporting this, we measured significant increases in coherence between the hippocampus and TT when optogenetically activating LINCs, across a range of stimulation frequencies (mixed-design ANOVA with Greenhouse-Geisser correction: genotype p=0.04, F = 4.7, DF = 1; stimulation frequency*genotype p=0.40, F = 1.0, DF = 3; *Figure 9e*). Together, these data suggest that LINCs can impact hippocampal function, strongly entrain hippocampal oscillations, and increase coherence between the hippocampus and downstream regions.

## Discussion

We have taken advantage of recent advances in viral vector specificity (i.e. the INTRSECT approach; *Fenno et al., 2014*) to selectively label and manipulate a population of CA1 cells that previously lacked any detailed description: LINCs. In addition to this being the first time that LINCs have been described in any detail, this work revealed that LINCs have several properties that make them especially unique and exciting. (1) LINCs have widespread postsynaptic connections within the hippocampus, allowing them to provide both fast and long-lasting GABAergic inhibition to almost any cell in CA1. (2) LINCs have long-range projections to many distinct regions of the brain. (3) Manipulating LINCs can cause spatial memory deficits, indicating that LINCs can have a significant impact on hippocampal function. And (4) LINCs can drive oscillatory activity in the hippocampus and increase interregional coherence between the hippocampus and projection targets. In summary, LINCs are poised to have a significant impact on the network, and a detailed understanding of LINCs is therefore important for a proper understanding of the hippocampal formation, and its downstream connections, more broadly.

We found that LINCs are distinct from previously characterized GABAergic hippocampal nNOS cells, and are a somewhat heterogeneous population of cells. LINCs reside primarily in stratum oriens and stratum pyramidale, and have horizontally or vertically oriented, sparsely spiny, dendritic arbors. They display a regular spiking non-pyramidal firing pattern and rarely express somatostatin (SOM), parvalbumin (PV), vasoactive intestinal peptide (VIP), calretinin (CR), or the muscarinic acetylcholine receptor M2 (M2R). LINCs provide widespread, powerful, inhibition within the hippocampus, and, when optogenetically stimulated, are able to alter hippocampal function and oscillations. LINCs project to several areas outside of the hippocampus, including, but not limited to, the medial

septum, diagonal band, tenia tecta, olfactory tubercle, and retromammillary nucleus. Activation of LINCs is able to alter oscillatory activity in downstream regions, and increases interregional coherence. Altogether, LINCs, though somewhat heterogeneous, are unified in that they are GABAergic, have long-range projections, and express nNOS.

LINCs are not the only GABAergic cell population in the hippocampus that express nNOS, but can be distinguished from these other neuronal populations in a variety of ways. A subpopulation of neurogliaform cells express nNOS (*Krook-Magnuson et al., 2011*; *Tricoire et al., 2010*), but neurogliaform cells have a tight, dense, axonal plexus, and short and radially oriented thin dendrites (*Price et al., 2005*; *Tricoire et al., 2010*). Neurogliaform cells also display a late-spiking firing pattern with a high rate of persistent, or axonal-barrage (*Sheffield et al., 2013*), firing (*Armstrong et al., 2012*; *Krook-Magnuson et al., 2011*). By contrast, LINCs have broad dendrites and far-reaching axons, show a regular spiking non-pyramidal firing pattern, and do not have a high rate of axonal-barrage firing. Although LINCs are therefore distinct from nNOS-expressing neurogliaform cells, we were also able to label neurogliaform cells with our intersectional approach when the viral vector was injected instead near the stratum lacunosum moleculare/stratum radiatum area, where neurogliaform cells are especially dense (*Price et al., 2005*; *Tricoire et al., 2010*). It is likely that the small radius of neurogliaform cells' axonal and dendritic arbors limits uptake of the virus unless the injection is made in close proximity to the location of their cell bodies, and this in turn contributes to the labeling of LINCs (rather than neurogliaform cells) with our specific experimental approaches.

Another population of neurons that are reported to express nNOS are a subtype of interneuron selective interneurons (*Acsády et al., 1996*; *Gulyás et al., 1996*; *Tricoire et al., 2010*) that display complex or 'hooked' after-hyperpolarizations (*Tricoire et al., 2010*), and, importantly, express both VIP and CR (*Tricoire et al., 2010*; *Tricoire et al., 2011*). As very few eYFP-labeled cells in this study expressed either VIP or CR (and even fewer both; 8.8% were VIP+, 4.0% were CR+, and 3.3% were both VIP+ and CR+), it appears that our intersectional approach did not target nNOS+/VIP+/CR+ interneuron selective interneurons. Although the dendritic and axonal arbor of interneuron-selective interneurons is not as tight as that of neurogliaform cells, it is possible that the relative sparsity of the nNOS+ subtype of interneuron-selective interneurons prevented their noticeable labeling in our study; it has been reported that only 7% of all nNOS-positive CA1 interneurons are also VIP+ (*Tricoire et al., 2010*). It is also worth noting that the relative lack of VIP co-expression in LINCs additionally distinguishes LINCs from recently identified VIP+ CA1 inhibitory projection cells (which project to the subiculum; *Francavilla et al., 2018*).

Neocortical NOS-type I cells, like LINCs, have long-range projections and express nNOS (*Tomioka et al., 2005*; *Tricoire and Vitalis, 2012*), but the existence of hippocampal Type I-like cells has been somewhat controversial (*Fuentealba et al., 2010*; *Quattrocolo et al., 2017*). However, using gene expression profiling (*Harris et al., 2018*), a population of CA1 cells believed to correspond to 'back-projection' cells (*Klausberger, 2009*) have been identified as being similar to neocortical NOS-type I cells, because of their shared and consistently high levels of both SOM and NPY expression (*Sik et al., 1994*). Back-projecting cells target other hippocampal subfields including CA3 and the dentate gyrus (*Klausberger, 2009*; *Sik et al., 1994*). Although a small subset of LINCs may match back-projecting, NOS-type I-like cells, this would be a very small subset. Although 46.9% of CA1 eYFP+ cells expressed NPY, only 4.9% expressed SOM, and only 1.8% were immunopositive for both NPY and SOM. Similarly, we noted no axonal drumstick-like appendages (*Gulyás et al., 2003*; *Sik et al., 1994*) on LINCs. Therefore, few, if any, LINCs have characteristics reminiscent of back-projecting or NOS-type I-like cells. It is also worth noting that there may be differences in both expression profiles based on gene expression in dissociated cells and expression profiles based on immunohistochemistry in fixed tissue. These differences can stem from changes in expression during the dissociation process as well as from differing levels of sensitivity. Even within immunohistochemical profiles, sizeable differences in labeling can result from differing protocols, different activity levels, and dendritic versus somatic expression (*Figure 1—figure supplement 1*). However, the use of immunohistochemical profiles best allowed us to compare LINCs to other hippocampal populations, as the vast majority of previous literature describing hippocampal populations has used immunohistochemistry-based approaches to label neurons.

An additional previously identified population of GABAergic CA1 cells with extrahippocampal projections are hippocampal-septal (or double-projection; *Jinno et al., 2007*) cells. Like LINCs, these

cells reside in the stratum oriens of CA1 (*Alonso and Köhler, 1982*; *Deller and Leranth, 1990*; *Freund and Buzsáki, 1996*; *Gulyás et al., 2003*; *Jinno et al., 2007*; *Mattis et al., 2014*; *Villette et al., 2016*), and, as the name suggests, they project to the medial septum/diagonal band (*Alonso and Köhler, 1982*; *Toth et al., 1993*). The vast majority (in our hands, ≥80% [data not shown]) of hippocampal-septal cells express SOM (*Jinno et al., 2007*; *Villette et al., 2016*). As LINCs project to the medial septum, and some eYFP+ cells that were co-labeled with Fluorogold after injection into the medial septum were also SOM+, it is likely that there is some overlap between LINCs and previously identified hippocampal-septal cells. However, as LINCs robustly project to areas that SOM+ hippocampal-septal cells do not (e.g. the tenia tecta; *Figure 4—figure supplement 2*), and as the vast majority of LINCs do not express SOM (*Figure 5*), the overlap between LINCs and SOM+ hippocampal-septal cells is limited (~5% of LINCs are SOM+).

Other described hippocampal GABAergic projection neurons include trilaminar cells (which reside in the oriens of CA1, locally have axons that target stratum oriens, pyramidale, and radiatum, and have long-range projections to the subiculum and possibly other cortical areas) and oriens-retro-hippocampal cells (which also reside in the oriens of CA1 and project to 'retrohippocampal' areas, which in this context include the subiculum, presubiculum, parasubiculum, entorhinal cortex, retrosplenial cortex, and/or indusium griseum) (*Jinno et al., 2007*; *Klausberger, 2009*). Any overlap between these cells and LINCs appears to be minimal, as these cells typically strongly express the muscarinic acetylcholine receptor M2 (M2R) (*Jinno et al., 2007*; *Klausberger, 2009*), and very few LINCs are M2R immunopositive.

Previously, hippocampal non-pyramidal neurons, including in CA1, with projections to the supramammillary region were noted in cat tissue (*Ino et al., 1988*), but their identity was otherwise entirely unknown. Our data suggest that at least some of these observed non-pyramidal projection neurons may have been LINCs.

We and others have reported that a percentage of hippocampal PV+ cells can also have long-range projections (*Christenson Wick et al., 2017*; *Eyre and Bartos, 2019*; *Goodman and Sloviter, 1992*). Specifically, sparse PV commissural connections to the contralateral hippocampal formation have been described (*Christenson Wick et al., 2017*; *Eyre and Bartos, 2019*). LINCs, too, appear to have sparse commissural connections (*Figure 4—figure supplement 1*). However, the majority of LINCs were PV-immunonegative, again arguing for limited overlap with this previously described cell group. LINCs are further distinguishable from PV cells in a number of ways. (1) LINCs display a regular spiking non-pyramidal firing pattern, whereas PV cells are fast spiking with tight action potential widths. (2) Optogenetic activation of LINCs can produce a postsynaptic GABA$_B$ response, whereas optogenetic activation of PV cells with similar experimental conditions produces a pure GABA$_A$ response (data not shown). Note that while this distinguishes LINCs from PV cells, a postsynaptic GABA$_B$ response following the optogenetic activation of other cell types has been reported (*Nichol et al., 2018*), and neurogliaform cells are able to produce a postsynaptic GABA$_B$ response following a single action potential in a single neurogliaform cell (*Oláh et al., 2007*; *Price et al., 2008*; *Tamás, 2003*). (3) PV cells preferentially target dPCs (*Lee et al., 2014*; *Valero et al., 2015*), whereas LINCs appear to target both dPCs and sPCs equally. This would place LINCs in a position where they are potentially able to coordinate activity between these two information streams (*Krook-Magnuson et al., 2012*; *Valero and de la Prida, 2018*) in a way that PV cells could not. It is important to note, however, that the apparent equal targeting of dPCs and sPCs by LINCs was observed with optogenetic manipulation while recording from PCs at varying anterior-posterior and proximal-distal positions. The increased variability introduced by this diverse sampling may have masked subtle differences in targeting (*Cembrowski and Spruston, 2019*). Similarly, optogenetically activating LINCs (as a population) may have masked potential LINC-subpopulation target selectivity. However, optogenetic activation of PV cells was reported to allow detection of the selective targeting of dPCs by PV cells (*Lee et al., 2014*). Therefore, it appears that LINCs provide broader inhibition within CA1 than PV cells.

The broad local targeting by LINCs additionally highlights potential differences between LINCs and some other long-range inhibitory projection neurons. Specifically, CA1 VIP+ neurons that project to the subiculum locally target interneurons selectively in CA1 (*Francavilla et al., 2018*). Similarly, interneurons, rather than pyramidal cells, were reported to be the hippocampal targets of hippocampal-septal cells on the basis of electron microscopy analysis (*Gulyás et al., 2003*). However, other reports suggest that hippocampal-septal cells (*Takács et al., 2008*), double-projection

cells (which are hippocampal-septal cells with additional projections to the subiculum) (*Jinno et al., 2007*), and oriens-retrohippocampal cells (*Jinno et al., 2007*) predominantly target pyramidal cells locally instead. Trilaminar cells are reported to preferentially innervate interneurons in CA1, but also to target pyramidal cells (*Ferraguti et al., 2005*). Our results suggest that, collectively, LINCs broadly target CA1 pyramidal cells *and* CA1 inhibitory neurons alike. As LINCs target CA1 pyramidal cells and inhibitory neurons, they are in a position to both inhibit pyramidal cells directly and potentially to disinhibit pyramidal cells (via inhibition of inhibition).

High postsynaptic connectivity and long-range projections are reminiscent of early-generated (EG), GABAergic hub cells, which are capable of orchestrating network-wide synchronous activity (*Bonifazi et al., 2009*; *Picardo et al., 2011*; *Villette et al., 2016*). Similar to LINCs, hub cells are unified by their widespread axonal arborization, but they display some morphological heterogeneity in both axonal structure (i.e., some hub cells are perisomatic targeting [compare to LINC in *Figure 2c*] whereas others have dendritically targeting axons [compare to LINC in *Figure 2b*]) (*Bonifazi et al., 2009*) and dendritic morphology (including cells with largely horizontal or largely vertical dendrites) (*Picardo et al., 2011*). In addition, both EG GABAergic hub cells and LINCs have broad hippocampal and extrahippocampal targets. However, LINCs also have notable differences when compared to EG hub cells, including electrophysiological properties (recorded EG cells had irregular/stuttering or burst adapting firing patterns) and expression levels of SOM (prevalent in EG hub cells) and nNOS (uncommon in EG hub cells) (*Picardo et al., 2011*).In addition, EG GABAergic hub cells are reported to be generated before E10.5 (*Picardo et al., 2011*), whereas BrdU labeling of LINCs peaked around E11 (*Figure 6*). In summary, while LINCs have features that are reminiscent of other hippocampal GABAergic cells, no previously described cell population adequately captures their collective identity.

Given the extensive prior examination of inhibitory neurons in CA1 (*Freund and Buzsáki, 1996*; *Klausberger and Somogyi, 2008*), it seems surprising that any cell population, especially one with such widespread connections as LINCs, would have evaded prior characterization. In this regard, it is important to consider that nNOS-expressing cells in the SO and SP with dendrites suggestive of LINCs have indeed been noted (*Freund and Buzsáki, 1996*), but that further investigation was hampered. Many different factors have probably contributed to the prior difficulty in studying these cells. First, nNOS immunohistochemistry is notoriously challenging (*Burette et al., 2002*), and LINCs can express relatively low levels of nNOS, as well as dendritically concentrated nNOS (*Burette et al., 2002*), which further complicates easy detection (*Figure 1—figure supplement 1*). Moreover, we found that other common long-range projection molecular markers are insufficient for labeling LINCs (*Figure 5*). Similarly, although NADPH-d staining was previously able to identify axon fragments in the fimbria, the reaction was unable to label axons fully, and therefore their sources and trajectories could not be determined (*Higo et al., 2009*). In addition, as nNOS is expressed in other CA1 populations, identifying LINCs on the basis of immunohistochemistry alone becomes extremely difficult, as the morphology may not be sufficiently visible. Indeed, as even pyramidal cells express nNOS (*Burette et al., 2002*), taking a simple nNOS-Cre based approach to target LINCs transgenically or virally would be insufficient. Consequently, the recently developed intersectional approach (*Fenno et al., 2014*) was key. Our selective labeling of LINCs was also due to a degree of serendipity, as other interneuron populations that also express nNOS have more restricted processes (*Armstrong et al., 2012*; *Freund and Buzsáki, 1996*; *Tricoire and Vitalis, 2012*). Similarly, retrograde-based labeling or expression systems suffer from the relative rarity of LINCs and the fact that many of the areas targeted by LINCs are also targeted by other cell populations (*Arszovszki et al., 2014*; *Cenquizca and Swanson, 2007*), which may have overwhelmed the ability to identify LINCs previously, especially those residing in the stratum pyramidale (*Gulyás et al., 2003*). Although significant trouble shooting may be required, the labeling of LINCs in non-transgenic animals (including wildtype mice, rats, or other species) may be possible in future work, for example through a combination of the recently developed vectors that use the Dlx enhancer to limit expression to inhibitory neurons (*Dimidschstein et al., 2016*) and an additional approach to limit expression to, for example, cells that project to the tenia tecta. In light of the numerous complications in selectively labeling and identifying LINCs, it is not entirely surprisingly that LINCs have largely evaded previous consideration.

To begin to explore whether LINCs could have a functional impact on hippocampal function, we optogenetically activated LINCs, including during the ORM and OLM tasks. Disrupting the native

activity patterns of LINCs in this way (via synchronous activation at ~7 Hz) altered on-going theta oscillations in the hippocampus and impaired performance on the spatial OLM task. Notably, we manipulated LINCs during both encoding and retrieval portions of the OLM, and therefore we are unable to discern whether these effects were due to impairments in encoding and/or to impairments in retrieval specifically. However, our data do illustrate that the impairment in the OLM task was not due to general indifference to the objects or, for example, to motor deficits, because a similar amount of time was spent exploring the objects in total (there was no preferential investigation of the object in the novel location). Development of novel tools (e.g., an intersectional version of an inhibitory opsin) will allow us in the future to examine the impact of the selective silencing (rather than activation) of LINCs on hippocampal oscillations and performance of spatial tasks. Future work will also answer important additional questions, including which cells are targeted by LINCs in downstream areas, what excitatory, inhibitory, and modulatory input do LINCs receive, and when are LINCs natively active.

To further explore the ability of LINCs to alter hippocampal oscillations, we stimulated LINCs at a range of frequencies. We found that optogenetic manipulation of hippocampal LINCs not only altered hippocampal oscillatory activity across a wide range of frequencies, but also altered oscillatory activity in the tenia tecta, a forebrain region receiving input from LINCs. Moreover, we found an increase in coherence between the tenia tecta and the hippocampus during optogenetic activation of LINCs. Increased synchrony or coherence between brain regions can be task-induced (*Hoffmann and Berry, 2009*; *Wikgren et al., 2010*; *Yu and Krook-Magnuson, 2015*), is believed to allow coordinated activity and efficient information transfer between regions (*Buzsáki et al., 2013*; *Colgin, 2011*; *Fries, 2005*; *Singer, 1999*; *Sirota et al., 2008*), including between the dorsal hippocampus and prefrontal cortex (*Hyman et al., 2010*; *Jones and Wilson, 2005*), and can correlate with memory and task performance (*Hyman et al., 2010*; *Jones and Wilson, 2005*; *Spellman et al., 2015*). Our findings further support the idea that long-range inhibitory neurons, such as LINCs, may help to coordinate activity between brain regions (*Jinno, 2009*).

Taken together, our data clearly indicate that LINCs should no longer be overlooked. LINCs have a broad impact on the hippocampus, and appear to play a role in coordinating hippocampal and extrahippocampal activity. Future work will provide important additional information about the roles of LINCs not only in healthy physiology, but also pathophysiology, including determining whether manipulation of LINCs could provide therapeutic benefit, for example, in Alzheimer's disease (*Koliatsos et al., 2006*) or temporal lobe epilepsy (*Krook-Magnuson et al., 2013*).

## Materials and methods

All experimental protocols were approved by the University of Minnesota's Institutional Animal Care and Use Committee.

### Animals

Mice were sexed on the basis of their external genitalia at the time of weaning (postnatal day 21). Male and female mice were used in experiments; no significant differences were noted between males and females regarding the electrophysiological properties of LINCs, post-synaptic responses, discrimination indices, nor changes in power or coherence, and thus, data from males and females have been combined throughout.

For all experiments, mice were bred in-house. Dlx5/6-Flpe founders, expressing Flpe recombinase under the *Dlx5/Dlx6* (*id6/id5*) intergenic enhancer region and β-globin basal promoter, were kindly provided by the Fishell lab (also available from Jackson Laboratory; Tg(ml56i-flpe)39Fsh/J, stock 010815, maintained as hemizygotes; *Miyoshi et al., 2010*). We refer to these mice, which express Flpe in GABAergic forebrain neurons, as 'fDLX'. nNOS-Cre founders, expressing Cre recombinase in nNOS-expressing neurons, were purchased from Jackson Laboratory (B6.129-*Nos1*tm1(cre)Mgmj/J; stock 017526; maintained as homozygotes; *Leshan et al., 2012*). We refer to these mice as 'cNOS', to denote the expression of Cre in nNOS-expressing cells. fDLX mice were crossed with cNOS mice to produce cNOS-fDLX mice, which were used for experiments. Positive offspring expressed both Cre and Flp recombinases in nNOS-expressing GABAergic neurons of the forebrain; Flpe-negative littermates were used as controls. In addition, uncrossed Flpe+ fDLX mice (n = 2) and Black6 mice (n = 2, C57BL/6J, Jackson Laboratory stock 000664) were injected with virus to further confirm

the specificity of viral expression in the absence of Cre. Reporter mice for Cre+Flpe (RCE:dual) were crossed with cNOS-fDLX mice to further examine the profile of Cre and Flpe expression in cNOS-fDLX mice. RCE:dual founder mice were also kindly provided by the Fishell lab (also available from Jackson Laboratory: Gt(ROSA)26Sor^tm1CAG-EGFP)Fsh/Mmjax, stock 32036-JAX, maintained as hemizygotes; *Miyoshi et al., 2010*).

For comparisons of projections to SOM-expressing inhibitory cells, mice expressing Cre recombinase in SOM+ neurons (cSOM; Ssttm2.1(cre)Zjh/J; stock 013044; maintained as homozygotes; *Taniguchi et al., 2011*) were crossed to fDLX animals. Positive offspring expressing both Cre and Flp recombinases were injected with virus and used for the quantification of neurites in the tenia tecta, medial septum, and horizontal limb of the diagonal band. In addition to the previously listed viral controls, a Flpe-negative cSOM-fDLX littermate was also injected with virus and used as a negative control; no expression was seen in this animal.

Except following implantation, animals were housed in standard housing conditions (12 hr light, 12 hr dark) in the animal facility at the University of Minnesota. Following implantation for behavioral experiments and in vivo electrophysiology, animals were singly housed in investigator-managed housing. In all housing conditions, animals were allowed ad libitum access to food and water.

## Stereotaxic surgeries

Surgical procedures were performed stereotaxically under 1–3% isoflurane anesthesia (*Armstrong et al., 2013*; *Christenson Wick et al., 2017*; *Krook-Magnuson et al., 2013*; *Zeidler et al., 2018*).

### Viral and Fluorogold injections

Mice were injected with 1 μL of a virus encoding the excitatory opsin channelrhodopsin (hChR2 (H134R); ChR2) fused to enhanced yellow fluorescent protein (eYFP) in a Cre- and Flp-dependent manner (AAV-DJ-hSyn-Con/Fon-hChR2(H134R)-eYFP-WPRE; *Fenno et al., 2014*; UNC Viral Vector Core lot numbers AV6214 and AV6214C, titer $4.4 \times 10^{12}$ vg/mL; Stanford Viral Vector Core lot numbers 1599, 3214, titers $2.0 \times 10^{13}$ and $2.55 \times 10^{13}$ vg/mL, respectively, or, where noted, AAV5-hSyn-Con/Fon-hChR2(H134R)-eYFP-WPRE; UNC Viral Vector Core lot number AV6149B, titer $2.3 \times 10^{12}$ vg/mL) via Hamilton syringe (model 7002KH) into the left dorsal hippocampus (0.2 cm posterior, 0.125 cm left, and 0.125 cm ventral to bregma) at an approximate rate of 200 nL/min at postnatal day 45 or greater. Additional cNOS-fDLX mice received viral injections instead targeting either the stratum oriens of CA1 in the ventral hippocampus (0.36 cm posterior, 0.28 cm left, and 0.28 cm ventral to bregma) or the stratum lacunosum moleculare of CA1 in the dorsal hippocampus (0.2 cm posterior, 0.125 cm left, 0.175 cm ventral to bregma). After the full volume of virus was injected, the syringe was left in place for at least 5 min before being withdrawn. Animals recovered on a heating pad and were returned to the animal housing facility the following day. Experiments were conducted at least 6 weeks following viral injection. Fluorogold (FG; 100 nL, 4% in saline; Fluorochrome LLC, cat#52–9400) was injected into the tenia tecta (0.22 cm anterior, 0.025 cm left, and 0.375 cm ventral to bregma), the vertical limb of the diagonal band (0.1 cm anterior and 0.5 cm ventral to bregma), or medial septum (0.075 cm anterior and 0.4 cm ventral to bregma). For each FG injection site, one FG-injected brain was harvested acutely to confirm targeting of the tracer injection (n = 3 acute brains). The remaining FG-injected brains (n = 3 tenia tecta, three diagonal band, three medial septum) were harvested 1 week after FG injection.

### Implant surgery

Mice used for behavioral and in vivo electrophysiology experiments were additionally implanted with a twisted-wire bipolar electrode (PlasticsOne, 2-channel stainless steel, MS303/3-A/SPC) (*Armstrong et al., 2013*) and optical fiber (Thorlabs, FT200UMT, Ø200μm, 0.39 NA) in the dorsal hippocampus near the injection site (0.2 cm posterior, 0.125 cm left, and 0.15 cm ventral to bregma). A second twisted-wire bipolar electrode was implanted in the tenia tecta (0.22 cm anterior, 0.05 cm left, and 0.375 cm ventral to bregma). Experiments were conducted minimally 5 days following the implant surgery to allow time for recovery.

## Slice electrophysiology recordings

cNOS-fDLX mice previously injected with virus were deeply anesthetized with 5% isoflurane and their brains were dissected. Coronal or sagittal hippocampal sections were prepared in ice-cold sucrose solution and incubated at 36°C for ~1 hr before being adjusted to room temperature until recording. All recordings were done at physiological temperature in artificial cerebrospinal fluid (ACSF). The sucrose solution contained the following (in mM): 85 NaCl, 75 sucrose, 2.5 KCl, 25 glucose, 1.25 $NaH_2PO_4$, 4 $MgCl_2$, 0.5 $CaCl_2$, and 24 $NaHCO_3$. The ACSF solution contained (in mM): 2.5 KCl, 10 glucose, 126 NaCl, 1.25 $NaH_2PO_4$, 2 $MgCl_2$, 2 $CaCl_2$, and 26 $NaHCO_3$ (*Krook-Magnuson et al., 2011*).

Slices were visualized with an upright microscope (Nikon Eclipse FN1) equipped with a xenon light source for visualizing fluorescence and optogenetic experiments (Lambda DG-4Plus, Sutter Instrument Company, model PE300BFA) (*Krook-Magnuson et al., 2013*). Recordings were performed using pipettes (3–4 MΩ) filled with an intracellular solution that had a relatively high chloride concentration to record $GABA_A$-mediated currents, and was cesium-free, to allow recording of $GABA_B$–mediated currents. The intracellular solution contained the following (in mM): 90 potassium gluconate, 43.5 KCl, 1.8 NaCl, 1.7 $MgCl_2$, 0.05 EGTA, 10 HEPES, 2 Mg-ATP, 0.4 $Na_2$-GTP, 10 phosphocreatine, and 8 biocytin, pH 7.29, 290 mOsm (*Krook-Magnuson et al., 2011*).

LINCs were quickly identified for recordings on the basis of their eYFP fluorescence and were later confirmed to be opsin-expressing on the basis of their light response that was resistant to antagonists. When determining LINCs' postsynaptic targets, cells were pseudo-randomly patched and were post hoc confirmed to be eYFP-negative. After establishing whole-cell configuration, the resting membrane potential was immediately recorded ($V_{rest}$) and the firing patterns of the recorded cells were examined with repeated current steps (500 ms step duration, increasing from an initial −300 pA hyperpolarization, with 10 pA or 50 pA step sizes) from a resting membrane potential adjusted to be approximately −60 mV (*Krook-Magnuson et al., 2011*; *Nichol et al., 2018*).

Recorded cells were then voltage clamped at −60 mV and tested for a response to blue light (5 ms duration, 10 s intersweep interval) (*Nichol et al., 2018*) and the series resistance was monitored. The amplitude of the averaged postsynaptic response was measured at its peak ([i] for $GABA_A$ responses average time to peak: 6.3 ms after the light was applied, n = 37 cells from 36 sections, 12 animals; [ii] for $GABA_B$ responses: 134.0 ms after light, n = 18 cells from 18 sections, 9 animals). A successful postsynaptic $GABA_A$ response was defined as an inward current greater than or equal to 10 pA below baseline; a $GABA_B$ response was defined as an outward current greater than or equal to 10 pA above baseline. Once a postsynaptic response was recorded, the $GABA_A$ receptor antagonist gabazine (5 µM, Sigma cat#S106) (*Szabadics et al., 2010*) was bath applied, and if a $GABA_B$ receptor response remained, the $GABA_B$ receptor antagonist CGP 55845 (5 µM, Sigma cat#SML0594) (*Price et al., 2008*) was added to the bath. No post-synaptic response was ever evident after the application of CGP 55845.

After recordings, slices were fixed in 0.1M phosphate buffer with 4% paraformaldehyde for ~24 hr at 4°C. Biocytin filling was then revealed with Alexa Fluor 594-conjugated streptavidin (Jackson Immuno Research, 016-580-084, 1:500). Some sections were further processed for diaminobenzidine (DAB) and/or camera lucida morphological reconstructions (*Krook-Magnuson et al., 2011*). Cell identity (i.e. LINC, sPC, dPC, IN) was determined post hoc on the basis of firing patterns, morphology, cell body location, and presence or absence of eYFP fluorescence. In addition, the LINC presented in *Figure 2c* was post-hoc tested for parvalbumin immunoreactivity, using immunohistochemistry procedures similar to those described below.

Electrophysiological parameters were examined using a procedure that was similar to that reported previously (*Krook-Magnuson et al., 2011*; *Ma et al., 2006*; *Sheffield et al., 2013*; *Sheffield et al., 2011*). Specifically, the following definitions were used:

## Action potential half-width near threshold and at max firing frequency (in milliseconds)

Full-width half-max (defined as the width of the action potential at the point of half-maximum voltage deflection) averaged within a cell across all the action potentials in the near threshold sweep (defined as the sweep with the first current step that produced more than two action potentials) or

the max firing sweep (defined as the sweep with the first current step that produced the most action potentials).

### Adaptation ratio of the interspike interval at max firing frequency (dimensionless)

1 – (first interspike interval from the current sweep that elicited the max firing frequency/last interspike interval from the same sweep).

### Coefficient of variance of the interspike interval at max firing frequency (dimensionless)

Standard deviation/mean interspike interval of all action potentials occurring during the current step that elicited the maximum firing frequency.

### Firing frequency near threshold and at max firing (in Hz)

Firing frequency was calculated as the number of action potentials divided by the duration of the current step.

### Persistent firing (yes/no binary)

Persistent firing (also known as axonal barrage firing *Krook-Magnuson et al., 2011*; *Sheffield et al., 2013*; *Sheffield et al., 2011*) was tested by repeated current injections (of 300 pA or greater, step duration 500 ms, 50% duty cycle). If no persistent firing was observed by 100 repeated current injections, the cell was said to not exhibit persistent firing.

### Threshold voltage (in mV)

Threshold voltage was measured from the first action potential elicited by the smallest current injection, with the threshold point roughly corresponding to zero of the second differential of the voltage immediately preceding the action potential.

## Immunohistochemistry, tissue processing, and cell counting

With the exception of the LINC in *Figure 2c*, all immunohistochemistry profiles of LINCs were determined from tissue specifically processed for that purpose. Mice were heavily anesthetized with approximately 5% isoflurane and decapitated. The brains were dissected and drop-fixed in 4% paraformaldehyde for ~48 hr, or, for VIP/CR, NPY/SOM, and some nNOS immunohistochemistry (*Figure 1—figure supplement 1*), brains were drop-fixed in 1% paraformaldehyde for less than 24 hr (*Burette et al., 2002*). Using a vibratome (Leica VT1000S), 50 µm coronal or sagittal brain sections were collected in 0.1M phosphate buffer at room temperature. After fixation and sectioning, free-floating immunostaining was performed on every fourth section for either nNOS (rabbit anti-nNOS, Cayman Chemical, cat#160870, 1:1000), PV (rabbit anti-PV, Swant, PV27, 1:1000), SOM (rat anti-SOM, Millipore Sigma, MAB354, 1:250), or M2R (rabbit anti-M2R, EMD Millipore, AB5166, 1:1000), in red (goat anti-rabbit Alexa Fluor 594, Jackson Immuno Research, 111-585-003, 1:500; donkey anti-rat Alexa Fluor 594, Jackson Immuno Research, 712-585-153, 1:500). Similarly, multiplexed free-floating immunostaining was conducted on every fourth section for either VIP (rabbit anti-VIP, Immunostar, 20077, 1:500) in far-red (AlexaFluor 647 donkey anti-rabbit, Jackson ImmunoResearch, 711-605-152, 1:500) and CR (guinea pig anti-CR, Swant, CRgp7, 1:500) in red (AlexaFluor 594 donkey anti-guinea pig, Jackson ImmunoResearch, 706-585-148, 1:500) or NPY (goat anti-NPY, Novus Biologicals, NBP1-46535SS, 1:250) in red (AlexaFluor 594, donkey anti-goat, ThermoFisher Scientific, A-11058, 1:500) and SOM (rat anti-SOM, Millipore Sigma, MAB354, 1:250) in blue (DyLight 405 donkey anti-rat, Jackson ImmunoResearch, 712-175-150, 1:500). Sections were then mounted with Vectashield mounting media. Mounting media with DAPI (Vectashield Antifade Mounting Media with DAPI, H-1200) was used for all tissues except those used in Fluorogold experiments or NPY/SOM multiplexed immunostaining.

Sections were visualized with epifluorescence or conventional transmitted light microscopy (Leica DM2500). eYFP-positive cells in the hippocampus were counted manually in every fourth 50 µm section in all planes of focus. Once an eYFP-positive cell body was identified, its dendritic morphology

was noted (horizontal, vertical, intermediate, or other) on the brain atlas and its soma was checked for colocalization with immunofluorescence and/or the retrograde tracer FG. In some instances, nNOS immuno-colocalization was found outside the soma (i.e. in the dendrite) in cells with nNOS immuno-negative cell bodies (*Figure 1—figure supplement 1c*). Confocal imaging was performed on an Olympus FluoView FV1000 BX2 upright confocal microscope. Images in figures were adjusted for brightness and contrast, with all adjustments applied to the entire image and equally for controls.

## Quantification of extrahippocampal projections

Confocal images of eYFP labeled projection fibers were taken in extrahippocampal regions from cNOS-fDLX mice and cSOM-fDLX mice (one image taken per location from 50 µm sections from three cNOS-fDLX and three cSOM-fDLX animals) that received an intrahippocampal injection of AAV-DJ-hSyn-Con/Fon-hChR2(H134R)-eYFP-WPRE. Extrahippocampal regions that were imaged include the medial septum (viewed sagittally, 0.0 mm lateral from bregma), the diagonal band of Broca (coronal, 0.5 mm anterior from bregma), and the dorsal tenia tecta (sagittal, 0.25 mm lateral from bregma). Neurites within the imaged region of interest were measured using Fiji's Simple Neurite Tracer.

## BrdU birthdating

Pregnant dams received a single injection (200 mg/kg) (*Cameron and McKay, 2001*; *Wojtowicz and Kee, 2006*) of the 5′-bromodeoxyuridine (BrdU; 10 mg/mL in saline (*Picardo et al., 2011*); Sigma, B5002-100MG, lots HMBF8725V and HMBF4669V) at embryonic day (E) 9.5, 10.5, 11.5, or 13.5. Time of vaginal plug discovery was defined as E0.5. One dam was injected at each embryonic stage. Once the offspring (n = 2 E9.5, 2 E10.5, 3 E11.5. 3 E13.5 offspring) reached postnatal day 45, they were injected with the AAV-DJ-hSyn-Con/Fon-hChR2(H134R)-eYFP-WPRE virus targeted to dorsal CA1 and euthanized 6 weeks later. Brains were drop-fixed in 4% paraformaldehyde for ∼48 hr before being sectioned into 50 µm sections. Every fourth section was processed for eYFP (goat anti-GFP, Abcam ab6673, 1:1000; AlexaFluor 488 donkey anti-goat, Abcam ab150129, 1:500) and then post-fixed in 4% paraformaldehyde overnight before being processed for BrdU immunohistochemistry (*Boulanger et al., 2016*). Following the eYFP pre-stain and post-fixation, sections were denatured in 1M HCl for 30 min at 45°C, then incubated with the rabbit anti-BrdU (ThermoFisher PA5-32256, 1:1000) primary and AlexaFluor 594 goat anti-rabbit (Jackson ImmunoResearch, 111-585-003, 1:500) secondary antibodies (*Boulanger et al., 2016*; *Wojtowicz and Kee, 2006*). Sections were then mounted with Vectashield mounting media with DAPI and viewed for colocalization of the GFP and BrdU immunofluorescent signals.

## Tissue clearing and imaging

Animals (n = 2) were perfused with ice cold phosphate buffered saline (PBS) followed by ice cold 4% PFA fixative, trimmed of excess right hemisphere tissue, and incubated in 4% PFA at 4°C overnight before active clearing (*Chung et al., 2013*) with X-CLARITY (Logos Biosystems). Cleared samples were incubated in refractive index matched solution (RIMS) (*Yang et al., 2014*) modified to preserve eYFP fluorescence and were imaged on a Nikon A1R inverted confocal microscope with a 10x glycerol immersion objective (0.5 numerical aperture, 5.5 mm working distance) at the University of Minnesota's University Imaging Center.

## Behavioral experiments

Object Recognition Memory (ORM) and Objection Location Memory (OLM) testing was performed as previously described (*Leger et al., 2013*; *Vogel-Ciernia and Wood, 2014*; *Zeidler et al., 2018*) with minor modifications. In addition to the ORM and OLM tests, subjects underwent an additional task which was similar to the ORM, but odorants were presented on cotton swabs as a way to test odor recognition memory (OdorRM, modified after *Scott et al., 2013*). Odorants (1-octanol [0.1 µL/mL], Sigma-Aldrich 95446, floral/citrus smell), nutmeg (1 mg/mL, Target Market Pantry), and vanillin (0.2 mg/mL, Sigma-Aldrich V1104) were diluted in mineral oil and were counterbalanced across mice (time spent investigating during encoding — 1-octanol: 5.7 ± 2.3 s, nutmeg: 5.4 ± 2.2 s, vanillin: 5.6 ± 2.2 s [mean ± SD]). Each test took place over 3 days, with 4 days in between each round of

testing. The order of the tests was counterbalanced across cohorts. For all tests, mice were habituated to the testing arena for 10 min on the first day. For the OdorRM test, two cotton swabs dipped in mineral oil were present throughout habituation. On the following (training/encoding) day, two identical objects (or two cotton swabs scented with the same odorant) were introduced to the arena and mice were allowed to explore for 10 min. On the third (testing/retrieval) day, two objects (or cotton swabs) were again placed in the arenas and mice were allowed to explore for 5 min. For the OLM, the objects remained the same on the testing/retrieval day, and one of the objects was moved to a new location (*Figure 7*). For the ORM/OdorRM tests, the location of objects/odors was constant, but one of the objects/odors was replaced with a new object/odor (*Figure 7*, *Figure 7—figure supplement 1*).

A train of blue light (5.0 ± 2.8 mW, Plexon PlexBright blue LED #94002–002 and driver #51382) pulses at approximately 7 Hz frequency (50 ms on, 100 ms off; as in *Krook-Magnuson et al., 2013*) was delivered for 3 s through the optical patch cable to the dorsal hippocampus once every 30 s during the training and testing periods. Training and testing sessions were video-recorded and manually analyzed for time spent exploring each of the two objects/cotton swabs by an experimenter blinded to mouse genotype. Object/odor investigation time was defined in the videos as the period when the mouse's nose was within 1 cm of the object/cotton swab's edge and oriented towards the object/cotton swab, excluding the time when the mouse was engaged in non-investigative behaviors (e.g. grooming or digging). A discrimination index (DI; *Vogel-Ciernia and Wood, 2014*) was calculated for the testing day by subtracting the time spent investigating the familiar object/odor from the time spent investigating the novel object/odor and dividing the result by the total time investigating both objects/odors, and multiplying this number by 100. Animals with a DI of greater than ± 20 on the encoding day were excluded (n = 6 animals), as were animals that investigated the objects for less than 3 s (ORM/OLM, n = 1 animal) (*Vogel-Ciernia and Wood, 2014*; *Zeidler et al., 2018*) or 2 s (OdorRM, n = 5 animals) during either training or testing. Note that animals displayed relatively low investigation times during the OdorRM task (*Figure 7—figure supplement 1*), and the exclusion criteria for that task was adjusted to 2 s accordingly. In addition, two positive animals were excluded for a lack of expression of the virus and a further two animals were excluded for viral expression mis-targeted to the dentate gyrus rather than CA1. These same four animals excluded for lack of expression or viral mis-targeting were also excluded from in vivo electrophysiology experiments (see below).

## In vivo electrophysiology recordings

After behavioral testing, electrical patch cables were connected to the hippocampal and tenia tecta electrode pedestals. The electrical LFP signal (the local differential between the tips of the twisted wires of the electrode) was amplified 5,000–10,000 times (Brownlee Precision 410, Neurophase) (*Armstrong et al., 2013*; *Krook-Magnuson et al., 2013*). A series of blue light stimulations was delivered to the dorsal hippocampus. These stimulations included the stimulation parameters used during behavioral training/testing as well as frequencies ranging from 2 Hz to 100 Hz (delivered in randomly assigned order, 30 s intertrial interval, light pulse width 5 ms, except for 7 Hz stimulation which matched the light stimulation protocol from behavioral testing [50 ms pulse width]).

## Statistical analysis

Statistical analyses were performed in Matlab R2014b and 2018a (including the Matlab Statistical Toolbox), and OriginPro 2016. A *p* value of less than 0.05 was considered statistically significant. Data are shown as mean ± standard error of the mean (SEM) unless otherwise specified. Median is shown for postsynaptic responses, as the data were strongly not normally distributed. For statistical analyses, planned group sizes were based on a priori power analyses (*Faul et al., 2007*) using preliminary data, the literature, and previous experience for estimates of variability and effect sizes. Actual group sizes are a combination of these planned group sizes and exclusions, as outlined above. In some cases, extra data were available (e.g. LFP traces of responses to light at a given frequency) and were included (i.e. data were not excluded if our final sample size after exclusions exceeded our a priori planned sample size). To ensure robust data sets when no statistical comparisons were made (e.g. immunohistochemistry), we collected data from three animals, counting cells

in every other or every fourth section across the entire anterior-posterior and medial-lateral axis of the hippocampus.

## Ex vivo electrophysiological recordings

The electrophysiological properties of LINCs were post-hoc compared across section orientation (n = 5 coronal, 18 sagittal), animal sex (n = 15 female, 8 male), virus injection location (n = 13 dorsal, 10 ventral/posterior), morphology of dendrites (n = 11 horizontal, 10 vertical; 2 cells' dendrites were not recovered and were excluded from this analysis), and cell body location (n = 16 in the stratum oriens/alveus, 7 in the stratum pyramidale) using Mann-Whitney tests. There were no significant differences between section orientation, sex, or virus injection location, and these variables were collapsed in further analyses. Potential differences between LINCs with different dendritic morphologies were noted (*Table 1* provides Bonferroni corrected alpha).

Postsynaptic response amplitudes were compared across cell types using a Kruskal-Wallis ANOVA (K-W) and post hoc two-tailed Mann-Whitney (M-W) tests when appropriate. Proportions of cell types showing postsynaptic $GABA_A$ and/or $GABA_B$ receptor-mediated responses and proportions of LINCs displaying persistent firing properties were compared using $\chi^2$ tests.

## Quantification of extrahippocampal projections

Total extrahippocampal projection neurite path lengths were divided by the volume of tissue imaged to determine the fiber length density per animal per region. Cumulative counts across animals were used for histograms of neurite lengths per region. The distribution of eYFP+ fiber lengths in the medial septum of cNOS-fDLX mice vs. that in cSOM-fDLX mice injected intrahippocampally with the intersectional Cre- and Flp-dependent virus were compared using a Kolmogorov-Smirnov (K-S) test.

## Behavioral testing

Retrieval discrimination indices (*Vogel-Ciernia and Wood, 2014*) ((investigation$_{novel}$ – investigation$_{familiar}$)/(investigation$_{novel}$ + investigation$_{familiar}$)*100) and total time spent investigating during retrieval were compared across genotype for each behavioral task using two-tailed Mann-Whitney tests.

## In vivo oscillations and coherence

In vivo electrophysiological recordings were analyzed in MatLab 2018a with custom software utilizing the Chronux library (Chronux 2.12) (*Mitra and Bokil, 2008*; *Mitra et al., 2018*). For each animal and each light-delivery frequency, traces (sampled at 1000 Hz) for both the hippocampal and the tenia tecta locations were aligned to onset of light delivery. The first step in analysis was to then remove traces that were likely to contain movement artifact: traces with a range (defined as the maximum recorded voltage value in the trace minus the minimum value) greater than two times the average range of all traces for that animal at that light-delivery frequency and electrode location were removed prior to further analysis, leaving on average approximately 110 traces for each animal at each of 26 different light-delivery frequencies.

In a subset of analyses, in order to evaluate how optogenetic ~7 Hz stimulation of LINCs impacted theta oscillations specifically during theta states (*Figure 8*), traces with a theta-to-delta ratio greater than 4.5 in the 3 s prior to light delivery were selected, with theta and delta ranges defined as 5–12 Hz and 1–3 Hz, respectively (*Buzsáki, 2002*; *Düzel et al., 2010*; *Gereke et al., 2018*). For visualization, these traces were bandpass filtered to the theta range (5–12 Hz) using Matlab's bandpass function.

To calculate the percent change in power at the stimulation frequency for each stimulation frequency for each animal, first the bandpower at the stimulation frequency (using a one hertz band centered around the stimulation frequency) was calculated for both the 3 s prior to light-delivery (i.e. baseline) and the 3 s during light delivery for each trial. These values were then averaged across trials, and the power during light was expressed as a percent increase over baseline for each mouse at each stimulation frequency by recording location (*Figure 9*, *Figure 9—figure supplement 1*).

The Chronux library was similarly used to calculate the trial averaged coherence between the two locations (hippocampus and tenia tecta), both for the 3 s prior to light delivery (baseline) and for the 3 s during light delivery (*Mitra and Bokil, 2008*; *Mitra et al., 2018*). Increase in mean coherence at

the stimulation frequency (±0.5 Hz) was then expressed as a percent increase from baseline (*Figure 9*, *Figure 9—figure supplement 1*).

To visualize the percent change in power across frequencies during light delivery per location, genotype, and stimulation frequency, differential (i.e., percent increase) spectrograms (*Figure 9—figure supplements 2–4*) were created as follows. For this analysis, stimulation frequencies plotted were limited to up to 40 Hz. Using Chronux, a trial averaged moving time spectogram was created for each animal/light-delivery combination for the 3 s prior to light delivery (baseline), using a moving window size of 1 s and a step size of 0.1 s. The calculated power per frequency bin was then averaged across the 3 s baseline time period. These averaged baseline values were then used to calculate the percent increase for the 3 s of light stimulation (again, using a 1 s window and 0.1 s step size). The resulting spectrograms were then averaged across animals according to genotype, by recording location (displayed in *Figure 9—figure supplements 2–4*).

## Custom software availability

Custom MATLAB code is available through GitHub at https://github.com/KM-Lab/SpectrumAndCoherence (*Krook-Magnuson, 2019*; copy archived at https://github.com/eLifeProduction/SpectrumAndCoherence).

## Acknowledgements

The authors would like to thank Gord Fishell and his lab, including Robert Machold, Goichi Miyoshi, and Rowena Turnbull, for providing the Dlx5/6-Flpe and RCE:dual reporter mouse lines, and Karl Deisseroth and Charu Ramakrishnan for the intersectional virus. We thank the University of Minnesota University Imaging Centers for tissue clearing, imaging and technical support, with special assistance from Director Mark Sanders. Additional thanks to the entire Krook-Magnuson lab, in particular to Casey Xamonthiene for expertise in tissue processing, Chris Krook-Magnuson for MATLAB programming assistance, Zachary Zeidler for assistance with behavior experimental design and confocal imaging, Zachary Montes and Isaac Hoff for assistance with behavioral experiments, Caara H Leintz for early assistance with stereotactic surgeries, and Bethany Stieve for helpful discussions and confocal imaging assistance. This work was supported by the National Institutes of Health grants R01-NS104071 (to EKM) and F31-NS105457 (to ZCW), the University of Minnesota's MnDRIVE (Minnesota's Discovery, Research, and Innovation Economy) initiative (to EKM and ZCW), and a McKnight Land-Grant Professorship (to EKM).

## Additional information

### Funding

| Funder | Grant reference number | Author |
| --- | --- | --- |
| National Institutes of Health | R01-NS104071 | Esther Krook-Magnuson |
| National Institutes of Health | F31-NS105457 | Zoé Christenson Wick |
| University of Minnesota | MnDRIVE (Minnesota's Discovery Research and Innovation Economy) initiative | Zoé Christenson Wick Esther Krook-Magnuson |
| University of Minnesota | McKnight Land-Grant Professorship | Esther Krook-Magnuson |

The funders had no role in study design, data collection and interpretation, or the decision to submit the work for publication.

### Author contributions

Zoé Christenson Wick, Conceptualization, Formal analysis, Investigation, Methodology, Writing—original draft, Writing—review and editing; Madison R Tetzlaff, Investigation, Methodology, Writing—review and editing; Esther Krook-Magnuson, Conceptualization, Formal analysis, Supervision, Funding acquisition, Methodology, Writing—review and editing

## Author ORCIDs
Zoé Christenson Wick https://orcid.org/0000-0002-2752-0140
Esther Krook-Magnuson https://orcid.org/0000-0002-6119-0165

## Ethics
Animal experimentation: All experimental protocols were performed in strict accordance with and approved by the University of Minnesota's Institutional Animal Care and Use Committee (protocol # 1801-35497A).

## Decision letter and Author response
Decision letter https://doi.org/10.7554/eLife.46816.031
Author response https://doi.org/10.7554/eLife.46816.032

# Additional files

## Supplementary files
• Transparent reporting form DOI: https://doi.org/10.7554/eLife.46816.029

## Data availability
All data used for analysis in this publication are included in the manuscript and supporting files. Source data has been included for Table 1. Custom MATLAB code is available through GitHub at https://github.com/KM-Lab/SpectrumAndCoherence (copy archived at https://github.com/eLifeProduction/SpectrumAndCoherence).

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
