## [Decision Letter]

Thank you for submitting your article "A novel population of long-range inhibitory neurons" for consideration by *eLife*. Your article has been reviewed by two peer reviewers, one of whom is a member our Board of Reviewing Editors, and the evaluation has been overseen by Laura Colgin as the Senior Editor. The reviewers have opted to remain anonymous.

The reviewers have discussed the reviews with one another and the Reviewing Editor has drafted this decision to help you prepare a revised submission.

Summary:

This paper describes a subpopulation of hippocampal GABAergic neurons which have an axon that targets extrahippocampal areas, a so-called long-range GABAergic neuron. These cells were discovered using intersectional approaches by injecting a Cre- and Flp- dependent virus carrying ChR2 expression in nNos-Cre x DLX-Flp mice. The authors note that their discovery was serendipitous, possibly related to the viral serotype and site of injection. They name these cells LINCs (long-range inhibitory nNOS-expressing cells) and study them morphologically, neurochemically and electrophysiologically (in vitro). By exploiting optogenetic approaches in vivo, they also characterize the behavioral and network effect of activating these cells with object recognition tasks and hippocampal/frontal lobe (tenia tecta) LFP recordings, respectively.

Major concerns:

1) It is not clear LINCs are a subtype of GABAergic neuron or a subtype of a subtype that is already known.

LINCS do not appear to fit a single category neither in morphological (vertical dendrites, horizontal, intermixed), neurochemical (N2R+, SOM+, PV+) and electrophysiologically. They seem to be a kind of hippocampo-septal cell but not only that, because they also project to the tenia tecta etc. According to single-cell transcriptomic data in Harris et al. PLoS Biol, Nos1+ cells are quite diverse. If authors feel this is a novel population, they should try to evaluate some of the novel classes identified by Harris et al., or maybe a combination of markers whenever possible.

Quantification of the population and innervation of regions such as the tenia tecta will help address the importance and definition of the LINCs.

Comparison of the measured endpoints to other GABAergic cell types would also be useful.

2) What is the explanation of the serendipity?

– Serendipity should be debugged to facilitate investigation by different labs in the future. For example, if serotype is the issue, what serotypes were tested. If site of injection is important, what coordinates? Is expression resulting from the interaction between the double transgenic animal (nNos-Cre x DLX-Flp) and the serotype or the promoter or the combination of both?

– Given that the expression of nNOS and SOM may be weak in the rodent hippocampus, one can imagine that mild transcriptional regulation by the intersectional approach may cause variable expression levels. The authors need to somehow test for this factor in their analysis. For instance, the relatively low expression levels of SOM, M2R and PV in LINCS (Figure 2B) suggests there should be a proportion of nNOS+ GABAergic negative for all these markers. Can the authors estimate similar numbers in WT mice with multiplexed immunostaining?

3) Numerous problems detract related to the presentation should be corrected. They relate to the order of presentation, the methods, or conceptual content.

4) Title: If proof that LINCs are a singular type of GABAergic neuron is not possible, the conclusions in the Title and elsewhere should be softened.

[Editors' note: further revisions were requested prior to acceptance, as described below.]

Thank you for resubmitting your work entitled "Novel long-range inhibitory nNOS-expressing hippocampal cells" for further consideration at *eLife*. Your revised article has been favorably evaluated by Laura Colgin (Senior Editor), a Reviewing Editor, and one reviewer.

The manuscript has been improved but there are three remaining issues that need to be addressed before acceptance, as outlined below:

1) Given morphological and neurochemical heterogeneity of LINCs, could the authors further clarify if they think LINCs are a new, separate type of nNOS+ GABAergic neuron or are a subtype?

2) Regarding the potential causes of the 'serendipitous" identification of LINCs, illustrated in Figure 2—figure supplement 2, the effect of the line (the DUAL reporter line) and AAV injections are not shown in Figure 2—figure supplement 2. These data should be added to Figure 2—figure supplement 2.

3) Regarding the idea to describe LINCs in wild type mice, can the authors either attempt it or could they discuss the issue and its associated difficulties in the text (i.e. difficulties of confirming LINC cell-types in WT mice)?

Reviewer #2:

Authors have addressed all my concerns. I would like to thank them for a thorough revision. The new narrative facilitates understanding and new data/analysis address my major points. In particular, authors provide additional immunostaining and birthdating data to support that LINCs may be a separate type of nNOS+ GABAergic cells. Whether they are a single population is still not clear (they are intrinsically heterogenous) but authors have softened the claim in title and text.

Another major point was overcoming serendipity of the discovery. The revised version directly address this point by evaluating the impact of the transgenic lines, viral serotype and injections. Authors argue that the intersectional approach is robust for simple AAV construct (Con/Fon) without the DJ serotype provided injection site is localized at the stratum oriens in their Cre/Flpe animals. Thus, the viral injection location appear a critical factor and it is illustrated in Figure 2—figure supplement 2, but neither the effect of the line (DUAL reporter line) nor AAV injections are shown. I would ask authors to add these data to Figure 2—figure supplement 2.

The only critical experiment they did not attempt was checking for putative LINCs in wild-type mice. Authors argue they will hardly separate them from neurogliaform cells based on multiplexed IH. However, they seem convinced about such a segregation in Figure 2—figure supplement 2 (blue are LINC, pink dots are neurogliaform) in terms of location, somatic size, etc.., which could inform debugging WT data. For instance, establishing similarities between somatic size, dendritic orientation and some markes can be helpful (based in Freund and Buzsaki description), even if not 100% confirmed. This is possibly the weakest part of the paper, as one can be left with the impression that LINC are just some by-product of the intersectional approach. Having said that, I still feel this paper is an important addition to the field even if those concerns remain. My only point is to highlight the issue in the text (i.e. difficulties of confirming LINC cell-types in WT mice).

---

## [Author Response]

Major concerns:1) It is not clear LINCs are a subtype of GABAergic neuron or a subtype of a subtype that is already known.

We have added new data, additional analysis, and have rearranged the text and expanded the Discussion section to clarify and better address how LINCs compare to previously described hippocampal cell populations.

LINCS do not appear to fit a single category neither in morphological (vertical dendrites, horizontal, intermixed), neurochemical (N2R+, SOM+, PV+) and electrophysiologically. They seem to be a kind of hippocampo-septal cell but not only that, because they also project to the tenia tecta etc. According to single-cell transcriptomic data in Harris et al. PLoS Biol, Nos1+ cells are quite diverse. If authors feel this is a novel population, they should try to evaluate some of the novel classes identified by Harris et al., or maybe a combination of markers whenever possible.

We have performed additional immunohistochemistry, including testing for LINC immunofluorescence with several additional markers and multiplexed immunostaining with combinations of markers to address this point. This data is provided in Figure 5.

Quantification of the population and innervation of regions such as the tenia tecta will help address the importance and definition of the LINCs.

We agree this is a valuable addition. Quantification of LINC fibers (as well as quantification of inhibitory projections from SOM+ hippocampal cells) in the septum, diagonal band, and tenia tecta are now included in Figure 4—figure supplement 2.

Comparison of the measured endpoints to other GABAergic cell types would also be useful.

To address this, we have rearranged the manuscript and expanded the Discussion section to better compare LINCs to previously described hippocampal cell populations including, but not limited to, neurogliaform cells, hippocampo-septal cells, cortical nNOS Type-1 cells, and interneuron-selective interneurons. We have also added birthdating experiments (Figure 6; allowing LINCs’ developmental timeline to be compared to early generated hub cells), quantification of inhibitory projections from SOM+ hippocampal cells (Figure 4—figure supplement 2; for comparison to LINCs’ extrahippocampal projections), and examination of the expression of SOM in LINCs that are Fluorogold positive after Fluorogold injection into the medial septum, diagonal band, or tenia tecta (Figure 4—figure supplement 2, panel E).

2) What is the explanation of the serendipity?– Serendipity should be debugged to facilitate investigation by different labs in the future. For example, if serotype is the issue, what serotypes were tested. If site of injection is important, what coordinates? Is expression resulting from the interaction between the double transgenic animal (nNos-Cre x DLX-Flp) and the serotype or the promoter or the combination of both?– Given that the expression of nNOS and SOM may be weak in the rodent hippocampus, one can imagine that mild transcriptional regulation by the intersectional approach may cause variable expression levels. The authors need to somehow test for this factor in their analysis. For instance, the relatively low expression levels of SOM, M2R and PV in LINCS (Figure 2B) suggests there should be a proportion of nNOS+ GABAergic negative for all these markers. Can the authors estimate similar numbers in WT mice with multiplexed immunostaining?

We have additionally tested for co-labeling of LINCs with other molecular markers including NPY, VIP, and CR and have added immunohistochemical multiplexing data (Figure 5). While nNOS cells with similar morphological properties to LINCs were previously described in wild type animals in Freund and Buzsaki’s *Interneurons of the Hippocampus* (see page 400 of that publication), it is not straight forward to estimate their numbers using multiplexing, as neurogliaform cells can also express both nNOS and NPY (and do not express e.g. SOM, PV, etc). As such, estimating LINC numbers based on immuno-multiplexing in wild type mice was not attempted. Instead, eYFP cell counts in cNOS-fDLX mice are provided in subsection “LINCs impact hippocampal function”.

To further explore the conditions required for labeling of LINCs vs neurogliaform cells with our intersectional genetic viral vector approach, we have explored 1) expression in our Cre and Flp lines, by crossing with a DUAL reporter line, 2) expression following AAV5 injection, rather than the DJ serotype we standardly used in our experiments, and 3) expression following injections in/near the stratum lacunosum moleculare, instead of the location we standardly used in our experiments (stratum oriens). Based on these, we find that 1) Cre and Flp expression matches what would be expected (as previously reported), 2) the DJ serotype is not a strict requirement (as AAV5 is able to label LINCs as well), and 3) neurogliaform cells are also able to be labeled (but appear to require that the viral vector be injected near their soma/processes). This information has been added to the manuscript.

3) Numerous problems detract related to the presentation should be corrected. They relate to the order of presentation, the methods, or conceptual content.

We have worked to address each of these minor points and to improve the overall presentation and flow of the manuscript.

4) Title: If proof that LINCs are a singular type of GABAergic neuron is not possible, the conclusions in the Title and elsewhere should be softened.

While many populations of neurons show heterogeneity, the reviewers’ point is well-taken. The title has been changed to avoid implying that LINCs are a singular type of GABAergic neuron, and to instead focus on the features that unify LINCs. The title is now “Novel long-range inhibitory nNOS-expressing hippocampal cells”.

[Editors' note: further revisions were requested prior to acceptance, as described below.]

The manuscript has been improved but there are three remaining issues that need to be addressed before acceptance, as outlined below:1) Given morphological and neurochemical heterogeneity of LINCs, could the authors further clarify if they think LINCs are a new, separate type of nNOS+ GABAergic neuron or are a subtype?

LINCs appear distinct from previously described hippocampal nNOS+ GABAergic neurons. This has been clarified in the second paragraph of the Discussion section.

2) Regarding the potential causes of the 'serendipitous" identification of LINCs, illustrated in Figure 2—figure supplement 2, the effect of the line (the DUAL reporter line) and AAV injections are not shown in Figure 2—figure supplement 2. These data should be added to Figure 2—figure supplement 2.

We have added these data to Figure 2—figure supplement 2.

3) Regarding the idea to describe LINCs in wild type mice, can the authors either attempt it or could they discuss the issue and its associated difficulties in the text (i.e. difficulties of confirming LINC cell-types in WT mice)?

We have expanded the Discussion section to include this issue.

Reviewer #2:

Authors have addressed all my concerns. I would like to thank them for a thorough revision. The new narrative facilitates understanding and new data/analysis address my major points. In particular, authors provide additional immunostaining and birthdating data to support that LINCs may be a separate type of nNOS+ GABAergic cells. Whether they are a single population is still not clear (they are intrinsically heterogenous) but authors have softened the claim in title and text.Another major point was overcoming serendipity of the discovery. The revised version directly address this point by evaluating the impact of the transgenic lines, viral serotype and injections. Authors argue that the intersctional approach is robust for simple AAV construct (Con/Fon) without the DJ serotype provided injection site is localized at the stratum oriens in their Cre/Flpe animals. Thus, the viral injection location appear a critical factor and it is illustrated in Figure 2—figure supplement 2, but neither the effect of the line (DUAL reporter line) nor AAV injections are shown. I would ask authors to add these data to Figure 2—figure supplement 2.

These images have been added to Figure 2—figure supplement 2.

The only critical experiment they did not attempt was checking for putative LINCs in wild-type mice. Authors argue they will hardly separate them from neurogliaform cells based on multiplexed IH. However, they seem convinced about such a segregation in Figure 2—figure supplement 2 (blue are LINC, pink dots are neurogliaform) in terms of location, somatic size, etc.., which could inform debugging WT data. For instance, establishing similarities between somatic size, dendritic orientation and some markes can be helpful (based in Freund and Buzsaki description), even if not 100% confirmed. This is possibly the weakest part of the paper, as one can be left with the impression that LINC are just some by-product of the intersectional approach. Having said that, I still feel this paper is an important addition to the field even if those concerns remain. My only point is to highlight the issue in the text (i.e. difficulties of confirming LINC cell-types in WT mice).

Identifying putative neurogliaform cells versus LINCs based on morphology is possible in virally labeled cells in Figure 2—figure supplement 2 because the morphology is generally appropriately visible. Unfortunately, immunohistochemistry with relevant markers does not allow sufficient visualization of the morphology necessary to distinguish between these neuronal populations. Selective labeling of LINCs in WT animals will require a combinatorial viral approach. The Discussion section has been expanded to further discuss this issue.